# Generating Diverse Cooperative Agents by Learning Incompatible Policies

**Rujikorn Charakorn**[1], **Poramate Manoonpong**[1,2], **Nat Dilokthanakul**[3]
[1]VISTEC, Rayong, Thailand   [2]SDU, Odense, Denmark   [3]KMITL, Bangkok, Thailand
`{rujikorn.c_s19, poramate.m}@vistec.ac.th, nat.di@kmitl.ac.th`

## Abstract

Training a robust cooperative agent requires diverse partner agents. However, obtaining those agents is difficult. Previous works aim to learn diverse behaviors by changing the state-action distribution of agents. But, without information about the task's goal, the diversified agents are not guided to find other important, albeit sub-optimal, solutions: the agents might learn only variations of the same solution. In this work, we propose to learn diverse behaviors via policy compatibility. Conceptually, policy compatibility measures whether policies of interest can coordinate effectively. We theoretically show that incompatible policies are not similar. Thus, policy compatibility—which has been used exclusively as a measure of robustness—can be used as a proxy for learning diverse behaviors. Then, we incorporate the proposed objective into a population-based training scheme to allow concurrent training of multiple agents. Additionally, we use state-action information to induce local variations of each policy. Empirically, the proposed method consistently discovers more solutions than baseline methods across various multi-goal cooperative environments. Finally, in multi-recipe Overcooked, we show that our method produces populations of behaviorally diverse agents, which enables generalist agents trained with such a population to be more robust.

## 1 Introduction

Cooperating with unseen agents (e.g., humans) in multi-agent systems is a challenging problem. Current state-of-the-art cooperative multi-agent reinforcement learning (MARL) techniques can produce highly competent agents in cooperative environments (Kuba et al., 2021; Yu et al., 2021). However, those agents are often overfitted to their training partners and cannot coordinate with unseen agents effectively (Carroll et al., 2019; Bard et al., 2020; Hu et al., 2020; Mahajan et al., 2022).

The problem of working with unseen partners, i.e., ad-hoc teamwork problem (Stone et al., 2010), has been tackled in many different ways (Albrecht & Stone, 2018; Carroll et al., 2019; Shih et al., 2020; Gu et al., 2021; Rahman et al., 2021; Zintgraf et al., 2021; He et al., 2022; Mirsky et al., 2022; Parekh et al., 2022). These methods allow an agent to learn how to coordinate with unseen agents and, sometimes, humans. However, the success of these methods depends on the quality of training partners; it has been shown that the diversity of training partners is crucial to the generalization of the agent (Charakorn et al., 2021; Knott et al., 2021; Strouse et al., 2021; McKee et al., 2022; Muglich et al., 2022). In spite of its importance, obtaining a diverse set of partners is still an open problem.

The simplest way to generate training partners is to use hand-crafted policies (Ghosh et al., 2020; Xie et al., 2021; Wang et al., 2022), domain-specific reward shaping (Leibo et al., 2021; Tang et al., 2021; Yu et al., 2023), or multiple runs of the self-play training process (Grover et al., 2018; Strouse et al., 2021). These methods, however, are not scalable nor guaranteed to produce diverse behaviors. Prior works propose techniques aiming to generate diverse agents by changing the state visitation and action distributions (Lucas & Allen, 2022), or joint trajectory distribution of the agents (Mahajan et al., 2019; Lupu et al., 2021). However, as discussed by Lupu et al. (2021), there is a potential drawback of using such information from trajectories to diversify the behaviors. Specifically, agents that make locally different decisions do not necessarily exhibit different high-level behaviors.

To avoid this potential pitfall, we propose an alternative approach for learning diverse behaviors using information about the task's objective via the expected return. In contrast to previous works that

use joint trajectory distribution to represent behavior, we use policy compatibility instead. Because cooperative environments commonly require all agents to *coordinate on the same solution*, if the agents have learned different solutions, they cannot coordinate effectively and, thus, are incompatible. Consequently, if an agent discovers a solution that is incompatible with all other agents in a population, then the solution must be unique relative to the population. Based on this reasoning, we introduce a simple but effective training objective that regularizes agents in a population to find solutions that are compatible with their partner agents but incompatible with others in the population. We call this method *"Learning Incompatible Policies"* (LIPO).

We theoretically show that optimizing the proposed objective will yield a distinct policy. Then, we extend the objective to a population-based training scheme that allows concurrent training of multiple policies. Additionally, we utilize a mutual information (MI) objective to diversify local behaviors of each policy. Empirically, without using any domain knowledge, LIPO can discover more solutions than previous methods under various multi-goal settings. To further study the effectiveness of LIPO in a complex environment, we present a multi-recipe variant of Overcooked and show that LIPO produces behaviorally diverse agents that prefer to complete different cooking recipes. Experimental results across three environments suggest that LIPO is robust to the state and action spaces, the reward structure, and the number of possible solutions. Finally, we find that training generalist agents with a diverse population produced by LIPO yields more robust agents than training with a less diverse baseline population. See our project page at https://bit.ly/marl-lipo

## 2 PRELIMINARIES

Our main focus lies in fully cooperative environments modeled as decentralized partially observable Markov decision processes (Dec-POMDP, Bernstein et al. (2002)). In this work, we start our investigation in the two-player variant. A two-player Dec-POMDP is defined by a tuple $(\mathcal{S}, \mathcal{A}^1, \mathcal{A}^2, \Omega^1, \Omega^2, T, O, r, \gamma, H)$, where $\mathcal{S}$ is the state space, $\mathcal{A} \equiv \mathcal{A}^1 \times \mathcal{A}^2$ and $\Omega \equiv \Omega^1 \times \Omega^2$ are the joint-action and joint-observation spaces of player 1 and player 2. The transition probability from state $s$ to $s'$ after taking a joint action $(a^1, a^2)$ is given by $T(s'|s, a^1, a^2)$. $O(o^1, o^2|s)$ is the conditional probability of observing a joint observation $(o^1, o^2)$ under state $s$. All players share a common reward function $r(s, a^1, a^2)$, $\gamma$ is the reward discount factor and $H$ is the horizon length.

Players, with potentially different observation and action spaces, are controlled by policy $\pi^1$ and $\pi^2$. At each timestep $t$, the players observe $o_t = (o_t^1, o_t^2) \sim O(o_t^1, o_t^2|s_t)$ under state $s_t \in \mathcal{S}$ and produce a joint action $a_t = (a_t^1, a_t^2) \in \mathcal{A}$ sampled from the joint policy $\pi(a_t|\tau_t) = \pi^1(a_t^1|\tau_t^1)\pi^2(a_t^2|\tau_t^2)$, where $\tau_t^1$ and $\tau_t^2$ contain a trajectory history until timestep $t$ from the perspective of each agent. All players receive a shared reward $r_t = r(s_t, a_t^1, a_t^2)$. The return of a joint trajectory $\tau = (o_0, a_0, r_0, ..., r_{H-1}, o_H) \in \mathcal{T} \equiv (\Omega \times \mathcal{A} \times \mathbb{R})^H$ can be written as $G(\tau) = \sum_{t=0}^{H} \gamma^t r_t$. The expected return of a joint policy $(\pi^1, \pi^2)$ is $J(\pi^1, \pi^2) = \mathbb{E}_{\tau \sim \rho(\pi^1, \pi^2)} G(\tau)$, where $\rho(\pi^1, \pi^2)$ is the distribution over trajectories of the joint policy $(\pi^1, \pi^2)$ and $P(\tau|\pi^1, \pi^2)$ is the probability of $\tau$ being sampled from a joint policy $(\pi^1, \pi^2)$.

We use subscripts to denote different joint policies and superscripts to refer to different player roles. For example, $\pi_A = (\pi_A^1, \pi_A^2)$ is a different joint policy from $\pi_B = (\pi_B^1, \pi_B^2)$, and $\pi_A^i$ and $\pi_A^j$ are policies of different roles.[1] Finally, we denote the expected joint return of self-play (SP) trajectories—where both policies are part of the same joint policy, $\pi_A$—as $J_{\text{SP}}(\pi_A) := J(\pi_A^1, \pi_A^2)$ and the expected joint return of cross-play (XP) trajectories—where policies are chosen from different joint policies, $\pi_A$ and $\pi_B$—as $J_{\text{XP}}(\pi_A, \pi_B) := J(\pi_A^1, \pi_B^2) + J(\pi_B^1, \pi_A^2)$.

Since we are interested in creating distinct policies for any Dec-POMDP, we need an environment-agnostic measure that captures the similarity of policies. First, we consider a measure that can compute the similarity between policies of the same role $i$, e.g., $\pi_A^i$ and $\pi_B^i$. We can measure this with the probability of a joint trajectory $\tau$ produced by either $\pi_A^i$ or $\pi_B^i$. However, in the two-player setting, we need to pair these policies with a reference policy $\pi_{\text{ref}}^j$. Specifically, $\pi_A^i$ and $\pi_B^i$ are considered similar if they are likely to produce the same trajectories when paired with an arbitrary reference policy $\pi_{\text{ref}}^j$. We define similar policies as follows:

---

[1] Note that LIPO can be applied to environments with more than two players with a slight modification. Specifically, a policy $\pi^j$ would represent the joint policy of all players except player $i$, $\pi^j(a_t^j|\tau_t^j) = \Pi_{k \neq i} \pi^k(a_t^k|\tau_t^k)$.

**Definition 2.1** (Similar policies). Considering two policies of the same role $i$, $\pi_A^i$ and $\pi_B^i$, and a reference policy $\pi_{\text{ref}}^j$ of a different role $j$, $\pi_A^i$ is similar to $\pi_B^i$ up to $\epsilon$ if and only if $\max_{\tau \in \mathcal{T}} |1 - \frac{P(\tau|\pi_A^i, \pi_{\text{ref}}^j)}{P(\tau|\pi_B^i, \pi_{\text{ref}}^j)}| \leq \epsilon$, where $0 \leq \epsilon \leq 1$.

Next, we consider an alternate view on assessing the similarity between policies using policy compatibility (Section 3). Policy compatibility measures the performance difference of a joint policy $\pi_B$ before and after one of its policies $\pi_B^i$ is substituted by another policy $\pi_A^i$. We define compatibility between a policy $\pi_A^i$ and a joint policy $\pi_B$ as follows:

**Definition 2.2** (Compatible policies). Given a policy $\pi_A^i$ and a joint policy $\pi_B$, $\pi_A^i$ is compatible with $\pi_B$ if and only if $J(\pi_A^i, \pi_B^j) \geq (1 - \epsilon)J_{\text{SP}}(\pi_B)$.

## 3 LEARNING INCOMPATIBLE POLICIES (LIPO)

Our goal is to create distinct policies and, therefore, a population of diverse agents. First, we theoretically show that policy compatibility can be used to identify whether two policies are different. Based on this observation, we propose a novel training objective that produces a distinct policy. Then, we extend this objective for training a population of diverse policies. Finally, we incorporate an MI objective that encourages each policy to learn local variations.

### 3.1 LEARNING A DISTINCT POLICY VIA POLICY COMPATIBILITY

In this section, we motivate our objective by looking at two joint policies: $\pi_A = (\pi_A^1, \pi_A^2)$ and $\pi_B = (\pi_B^1, \pi_B^2)$. The goal is for $\pi_A$ to learn a different behavior from $\pi_B$ via the compatibility criterion. Importantly, the compatibility criterion can be computed empirically without direct access to the trajectory distribution, which can be difficult to estimate. Under mild assumptions, we can simplify the setting such that a simple relationship between similarity measure and compatibility criterion emerges. By reasoning about the expected return under different pairs of policies, we derive our main result.

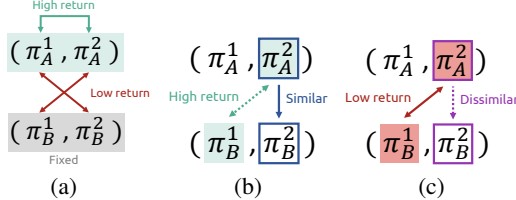

Figure 1: (a) The objective of $\pi_A$ (Eq. 1) in relation to $\pi_B$. (b, c) Conceptual illustration of Theorem 3.1 and Corollary 3.2. Solid lines represent given relationships, and dotted lines represent implied relationships.

**Theorem 3.1.** *If $\pi_A^i$ is similar to $\pi_B^i$, then $\pi_A^i$ is compatible with $\pi_B$. (The proof is in App. A.)*

**Corollary 3.2.** *If $\pi_A^i$ is not compatible with $\pi_B$, then $\pi_A^i$ is not similar to $\pi_B^i$.*

The result from Corollary 3.2 shows that we can find a policy $\pi_A^i$ that is not similar to $\pi_B^i$ by decreasing its compatibility with $\pi_B$ until they are incompatible, i.e., $J(\pi_A^i, \pi_B^j) < (1 - \epsilon)J_{\text{SP}}(\pi_B)$. Additionally, we can ensure that $\pi_A$ learns a meaningful solution by maximizing $J_{\text{SP}}(\pi_A)$. Assuming that $\pi_B$ has learned a solution and is fixed, the optimization objective of $\pi_A$ can be written as

$$\max_{\pi_A} \ J_{\text{SP}}(\pi_A) \ \text{subject to} \ J(\pi_A^i, \pi_B^j) < (1 - \epsilon)J_{\text{SP}}(\pi_B) \ \ \forall i, j \in \{1, 2\}, i \neq j \tag{1}$$

A way to solve such a constrained problem is to convert the constraints into regularization terms. For simplicity, we use a common $\lambda_{\text{XP}} > 0$ as a hyperparameter for the constraints. Then, we can write the soft objective of Eq. 1 as

$$\max_{\pi_A} J_{\text{SP}}(\pi_A) - \lambda_{\text{XP}} J_{\text{XP}}(\pi_A, \pi_B) \tag{2}$$

### 3.2 LEARNING A POPULATION OF DIVERSE POLICIES

To create a population of $N$ diverse policies, $\mathcal{P} = \{\pi_A | 1 \leq A \leq N\}$, we need an objective that requires each member of the population to have a different behavior relative to the rest of the population. We can write such an objective by expanding the XP term in Eq. (2) to include all other policies in the population. Additionally, we relax the assumption that other policies are fixed to

allow concurrent training of all policies. For a policy $\pi_A \in \mathcal{P}$, with an aggregation function $f_{\text{agg}}$, its objective becomes

$$\max_{\pi_A} J_{\text{LIPO}}(\pi_A, \mathcal{P}) = J_{\text{SP}}(\pi_A) - \lambda_{\text{XP}} \tilde{J}_{\text{XP}}(\pi_A, \mathcal{P}), \tag{3}$$

$$\text{where} \quad \tilde{J}_{\text{XP}}(\pi_A, \mathcal{P}) = f_{\text{agg}}(\mathcal{B}_A^{xp}), \tag{4}$$

$$\mathcal{B}_A^{xp} = \{ J_{\text{XP}}(\pi_A, \pi_B) \mid \pi_B \in \mathcal{P}_{-A} \}, \tag{5}$$

$$\mathcal{P}_{-A} = \mathcal{P} \backslash \{\pi_A\} \tag{6}$$

While using the average operation as the aggregation function is plausible, we find that using the max operation helps stabilize the training process and produces more diverse policies. We suspect that the average operation might produce many conflicting gradients and does not prioritize compatible XP pairs. We refer to $J_{\text{LIPO}}$ as the *compatibility gap* between a policy $\pi_A$ and a population $\mathcal{P}$.

We can see that the compatibility gap objective only uses the expected return ($J_{\text{SP}}$ and $\tilde{J}_{\text{XP}}$) and is insensitive to the state and action information. We argue that this distinction between LIPO and previous methods helps the agents discover more solutions in various situations (Sec. 4.1 and 4.5).

### 3.3 Inducing Variations in Each Policy

It is important to note that, regardless of the population size, there could be policies of role $i$ that are compatible with $\pi_A \in \mathcal{P}$ but not similar to $\pi_A^i$. We consider those policies to be variations of $\pi_A^i$ and propose to capture such variations via an MI objective. Specifically, we condition $\pi_A^i$ on a latent variable $z^i$ such that $\pi_A$ has the form of $\pi_A(a|\tau) = \mathbb{E}_{(z^1, z^2)} \pi_A^1(a^1|\tau^1, z^1)\pi_A^2(a^2|\tau^2, z^2)$ where $p(z^1, z^2)$ is a pre-defined prior distribution. We can induce variations of $\pi_A^i$ by maximizing $I(\{o^i, a^i\}; z^i)$, where $I(\cdot; \cdot)$ is the MI between two random variables. Intuitively, this objective encourages each policy to observe different observations and perform different actions given different values of the latent variable. However, maximizing $I(\{o^i, a^i\}; z^i)$ directly is intractable, instead we optimize the variational lower bound of the MI (Jordan et al., 1999) (see App. B for the derivation)

$$I(\{o^i, a^i\}; z^i) \geq H(z^i) + \mathbb{E}_{z^i, (o^i, a^i)}[\log q_{\phi_A}(z^i|o^i, a^i)], \tag{7}$$

where $q_{\phi_A}(z^i|o^i, a^i)$ is an approximation of the true posterior $p(z^i|o^i, a^i)$ parameterized by $\phi_A$. So, maximizing $I(\{o^1, a^1\}; z^1)$ and $I(\{o^2, a^2\}; z^2)$ is an optimization problem that can be written as

$$\max_{\pi_A, \phi_A} \frac{1}{2} \sum_{i=1}^{2} H(z^i) + \mathbb{E}_{z^i, (o^i, a^i)} \log q_{\phi_A}(z^i|o^i, a^i) \tag{8}$$

In the previous work (Mahajan et al., 2019), shared $z$ (i.e., $z^1 = z^2$) is used allowing both policies to *collectively* switch between different modes of behavior. However, LIPO uses independently sampled $z$ as it utilizes $z$ for a different purpose. Specifically, LIPO maximizes $J_{\text{LIPO}}$ to learn diverse solutions and optimizes the MI objective to learn variations of each solution. That is, the MI objective does not directly impact the diversity *between* different policies but increases variations of each individual policy. We note that the MI objective is optional; we show that without the MI objective, LIPO still produces diverse policies (Sec. 4.4).

### 3.4 Implementation

In practice, we modify the MI objective (Eq. 8) to be differentiable with respect to the policy $\pi_A^i$. Specifically, the variational posterior $q_{\phi_A}$ is modified such that, instead of a sampled action $a^i$, it takes the action distribution $\pi_A^i(\cdot|o^i, z^i)$ as an input, i.e., $q_{\phi_A}(z|o, \pi_A^i(\cdot|o^i, z^i))$. In contrast to previous MI-based approaches (Eysenbach et al., 2018; Sharma et al., 2019; Jiang & Lu, 2021; Lucas & Allen, 2022), we can optimize $I(\{o^i, a^i\}; z^i)$ directly without computing an auxiliary reward (Mahajan et al., 2019; Osa et al., 2022). The loss function of the modified MI objective is

$$L_{\text{MI}}(\pi_A, \phi_A) = -\frac{1}{2} \sum_{i=1}^{2} \mathbb{E}_{z^i, (o^i, a^i)} \log q_{\phi_A}(z^i|o^i, \pi_A^i(\cdot|o^i, z^i))) \tag{9}$$

The objective of a policy $\pi_A$ in a population $\mathcal{P}$ becomes

$$\max_{\pi_A, \phi_A} J_{\text{LIPO}}(\pi_A, \mathcal{P}) - \lambda_{\text{MI}} L_{\text{MI}}(\pi_A, \phi_A) \tag{10}$$

We set $z$ as a discrete variable and use the uniform distribution for $p(z^1)$ and $p(z^2)$. At the beginning of each episode, each policy is given an independently sampled $z$ that will be used until the end of the episode. We use MAPPO (Yu et al., 2021) for maximizing $J_{SP}$ and minimizing $\tilde{J}_{XP}$. More details, including the pseudocode and the extension to more than two players, can be found in App. D.

## 4 EXPERIMENTS

We study the effectiveness of LIPO under three multi-goal cooperative environments in which both players must collectively choose to accomplish one of the available goals. We evaluate the diversity of a population based on the number of distinct goals achieved. We compare LIPO to other cooperative MARL methods that do not require domain knowledge to generate diverse agents. Our baselines are as follows: (i) **Multi SP** (multiple runs of self-play), (ii) **SP$_{MI}$** (A single run of SP with added MI objective), (iii) **MAVEN** (Mahajan et al., 2019), and (iv) **TrajeDi** (Lupu et al., 2021). We also use **Multi SP$_{MI}$** and **Multi MAVEN** as baselines by training SP$_{MI}$ and MAVEN multiple times. We also discuss on methods that utilize domain knowledge in Sec. 5.

### 4.1 DISCOVERING DIVERSE SOLUTIONS

We use two simple environments to study the effectiveness of various methods in discovering solutions: (i) *One-Step Cooperative Matrix Game* (CMG), in which there are many possible solutions, and (ii) *Point Mass Rendezvous* (PMR), a temporally extended cooperative navigation environment.

**One-Step Cooperative Matrix Game (CMG):** A game of CMG is defined by a tuple $(M, \{k_m\}, \{r_m\})$, where $M$ is the number of solutions. For $m \in \{1, ..., M\}$, $k_m$ is the number of compatible actions and $r_m$ is the reward of a solution $m$. By choosing the same solution, both players get a reward $r_m$ associated with the chosen solution. We consider two setups of CMG: sub-optimal (CMG-S) and hard-to-find (CMG-H). For CMG-S, we set $(M = 32, k_m = 8, r_m = 0.5*(1+\frac{m-1}{M-1}))$, which causes each solution to have a different reward, ranging from 0.5 to 1. For CMG-H, we use $(M = 32, k_m = m, r_m = 1)$, which makes solutions with a smaller number of compatible actions harder to be found by random exploration. An example payoff matrix is shown in Fig. 2a.

**Point Mass Rendezvous (PMR):** The environment is based on the Multi-Agent Particle Environment (Lowe et al., 2017; Terry et al., 2020). The goal of this environment is for the two agents to navigate to a landmark together. There are $M = 4$ landmarks, and we consider each landmark as a solution in this environment. This environment has two modes: **PMR-C** and **PMR-L**. In PMR-C, landmarks are distributed evenly on the circumference of a circle. Thus, all landmarks are equally easy to find and optimal. In PMR-L, landmarks are placed on a line. In this scenario, closer landmarks are easier to find.

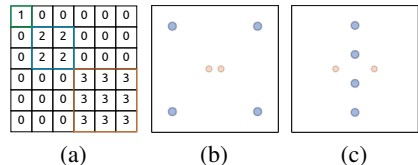

(a)       (b)       (c)

Figure 2: (a) The payoff matrix of a CMG game with $(M = 3, k_m = m, r_m = m)$. (b, c) The agents (orange) and landmark positions (blue) of PMR-C and PMR-L.

We define the population size $|\mathcal{P}|$ of each method as follows: For SP$_{MI}$ and MAVEN, $|\mathcal{P}|$ is equal to the number of dimensions of the latent variable, $|z|$. For Multi SP$_{MI}$ and Multi MAVEN, $|\mathcal{P}| = |z| \cdot n_{seed}$ where $n_{seed}$ is the number of random seeds and we use $|z| = 8$. For Multi SP, TrajeDi, and LIPO, $|\mathcal{P}|$ is the number of joint policies in the population.

**Results:** Fig. 3 shows the numbers of learned solutions, averaged over three runs. In all environments, LIPO consistently discovers more solutions than the baselines, given the same population size. The baselines find fewer solutions in CMG-H and PMR-L than they do in CMG-S and PMR-C, whereas LIPO performs similarly across settings. LIPO is also better than the baselines at finding sub-optimal solutions in CMG-S. We note that Multi SP and TrajeDi perform almost ideally in PMR-C, where all solutions are equivalent, but perform worse in other settings. Also, Multi SP$_{MI}$ finds all four solutions in PMR when the population size is bigger than 8. However, it performs poorly in CMG. LIPO's consistency across environments and settings demonstrates that LIPO is still effective when (i) many solutions exist, (ii) solutions are not equally optimal, and (iii) solutions are not equally likely to be found by random exploration. We also have experimented with stronger regularization coefficients for the baselines, which help the baselines discover more solutions. However, if the regularization coefficient is too large, they fail to produce capable policies.

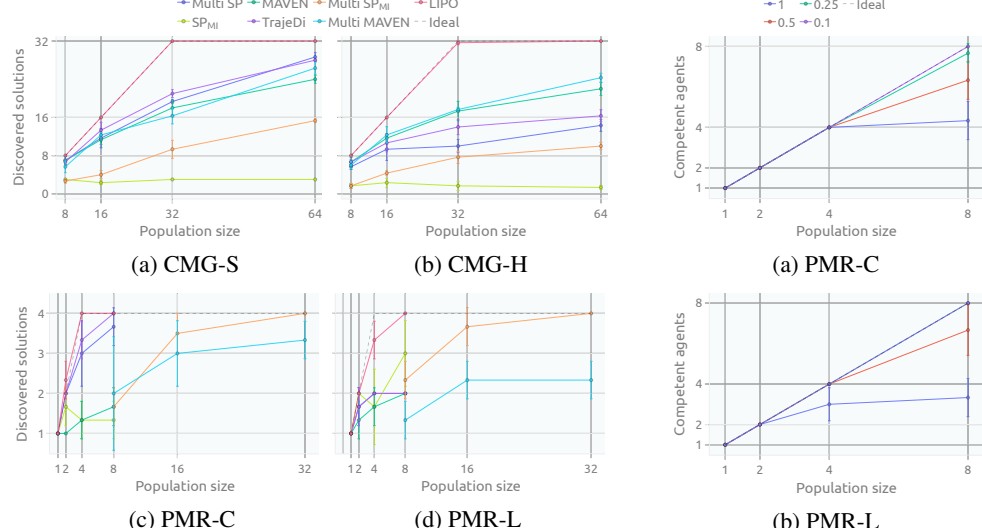

(a) CMG-S      (b) CMG-H      (a) PMR-C

(c) PMR-C      (d) PMR-L      (b) PMR-L

Figure 3: Numbers of discovered solutions. Ideally, if the population size increases by one, one more solution should be discovered, as depicted by the dashed lines (assuming that a joint policy does not produce a multi-modal behavior).

Figure 4: Numbers of competent joint policies using various combinations of N (x-axis) and $\lambda_{\text{XP}}$ (colors) in PMR.

## 4.2 TRADE-OFF BETWEEN COMPETENCY AND DISSIMILARITY OF JOINT POLICIES

It is possible that optimizing a regularized objective might incur training instability and create incapable policies. Here, we investigate the effect of different combinations of $\lambda_{\text{XP}}$ and the population size ($N$) on the competency of the policies.

Fig. 4 shows the number of competent joint policies when using different values of $N$ and $\lambda_{\text{XP}}$ in PMR. Particularly, in PMR, a joint policy is considered competent when both players stay close to a landmark at the end of an episode. We observe that when the population size is larger than the number of solutions ($N > M$), some surplus policies do not learn to reach a goal. Importantly, the number of competent joint policies depends on the value of $\lambda_{\text{XP}}$: lower values of $\lambda_{\text{XP}}$ yield more capable policies. However, using too low $\lambda_{\text{XP}}$ will generate policies that share a common solution when $N \leq M$ as shown in App. J.1.1. Additionally, when $N \leq M$, all trained agents are competent except when $\lambda_{\text{XP}}$ is too high in PMR-L. These results suggest that there is a trade-off between the number of capable joint policies and policy dissimilarity. When using a larger population size, a small $\lambda_{\text{XP}}$ should be used to avoid producing incompetent agents, while a bigger $\lambda_{\text{XP}}$ should be used with smaller population sizes to ensure the dissimilarity between joint policies.

## 4.3 TRADE-OFF BETWEEN COMPUTATION COST AND DIVERSITY

Not only is using bigger values of $N$ more likely to produce incompetent policies, but it is also computationally expensive. Formally, the computation complexity of approximating $\tilde{J}_{\text{XP}}(\cdot, \mathcal{P})$ is $\mathcal{O}(Nn_{xp})$ where $n_{xp}$ is the number of XP pairs used to approximate $\tilde{J}_{\text{XP}}(\pi_A, \mathcal{P})$. So, we investigate a way to reduce the cost of calculating $\tilde{J}_{\text{XP}}(\pi_A, \mathcal{P})$ by reducing $n_{xp}$. According to Eq. 5, the default value is $n_{xp} = N - 1$. When $n_{xp} < N - 1$, $n_{xp}$ policies are chosen randomly from $\mathcal{P}_{-A}$ by sampling without replacement.

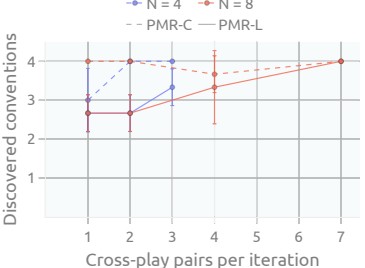

Figure 5: Number of learned solutions using various $n_{xp}$.

We observe that, while being computationally cheaper, using $n_{xp} < N - 1$ tends to produce less diverse populations as shown in Fig. 5. Thus, $n_{xp}$ can be considered a hyperparameter that controls the computation-diversity trade-off. However, as shown by the dashed lines, the effect of $n_{xp}$ on population diversity is less prominent in PMR-C, where solutions are equally likely to be found. We use $n_{xp} = N - 1$ in all other experiments. See App. J.1.2 for results in CMG.

## 4.4 Effect of the Mutual Information Objective

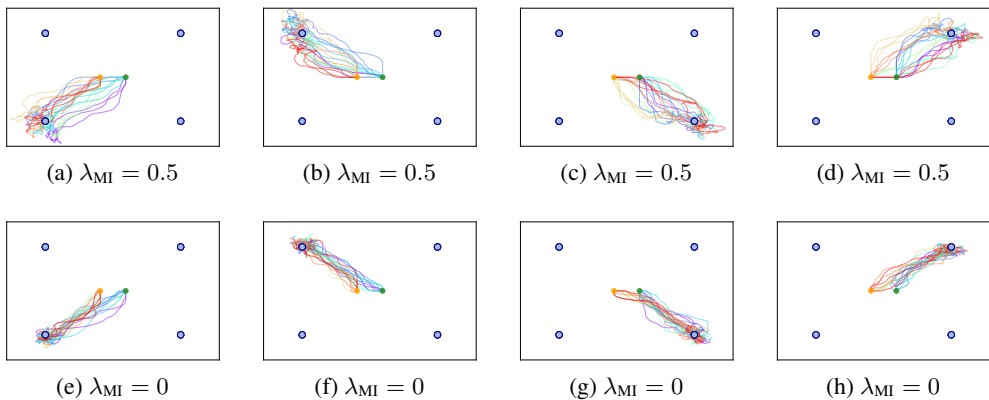

(a) $\lambda_{\mathrm{MI}} = 0.5$  (b) $\lambda_{\mathrm{MI}} = 0.5$  (c) $\lambda_{\mathrm{MI}} = 0.5$  (d) $\lambda_{\mathrm{MI}} = 0.5$

(e) $\lambda_{\mathrm{MI}} = 0$  (f) $\lambda_{\mathrm{MI}} = 0$  (g) $\lambda_{\mathrm{MI}} = 0$  (h) $\lambda_{\mathrm{MI}} = 0$

Figure 6: The top and bottom rows show four joint policies produced by a single run of LIPO training with and without the MI objective, respectively. Different colors of the trajectories correspond to different values of $z$. The orange and green circles show the starting positions. The blue circles represent the landmarks.

Fig. 6 shows the behaviors of the policies produced by LIPO with and without the MI objective in PMR-C. We can see the effect of the MI objective in the variety of the trajectories. Overall, each agent exhibits larger variations given a small MI regularization $\lambda_{\mathrm{MI}} = 0.5$. This result aligns with our motivation of using the MI objective to learn variations of each solution. With or without the MI regularization, LIPO discovers all the landmarks with $N = 4$.

## 4.5 Discovering Recipes in Multi-Recipe Overcooked

Overcooked, a collaborative cooking game, has been used to study the cooperative ability of learned agents in prior works (Carroll et al., 2019; Charakorn et al., 2020; Strouse et al., 2021; McKee et al., 2022). To investigate the usefulness of LIPO in a high-dimensional environment, we implement a more complex version of the game based on the work of Wu et al. (2021); players have to complete and serve one of the six pre-defined recipes as fast as possible, as opposed to delivering a single menu item re-

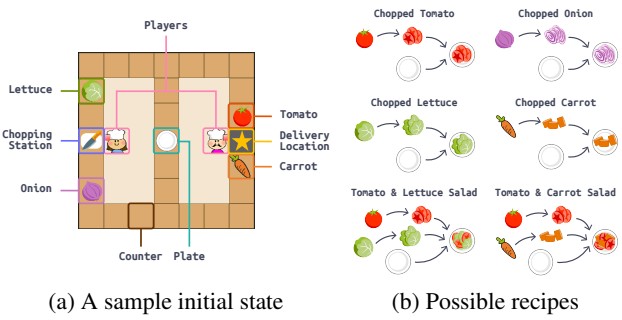

(a) A sample initial state    (b) Possible recipes

Figure 7: An overview of the multi-recipe Overcooked game.

peatedly. We emphasize that this environment is much more challenging than the ones in the previous experiments because of various aspects: First, it has a sparse reward signal. Second, there are multiple sub-tasks. Third, different recipes have different sub-tasks. Each of these characteristics of the environment complicates the process of finding diverse solutions. Furthermore, we note that recipes containing a carrot or a tomato are harder to complete than other recipes as they involve an additional coordination step. Particularly, carrot and tomato have to be sent over by the agent on the right, unlike lettuce and onion. Fig. 7 shows an overview of the game.

The goal in this experiment is to learn a population of behaviorally diverse agents. We choose to quantify the diversity of a population based on the entropy of its recipe distribution. For a population $\mathcal{P}$, we approximate the probability of recipe $i$ being completed as $P(\mathrm{recipe}_i | \mathcal{P}) \approx \frac{\sum_{\pi_A} m_i(\pi_A)}{\sum_i \sum_{\pi_A} m_i(\pi_A)}$, where $m_i(\pi_A)$ denotes the frequency of recipe $i$ under a joint policy $\pi_A$. The recipe frequencies, $\{m_i(\pi_A) | 1 \leq i \leq 6\}$, for each joint policy $\pi_A \in \mathcal{P}$ are measured by counting the completed recipes from 1,000 self-play episodes. For Multi SP, TrajeDi and LIPO, we set $N = 8$. For Multi SP$_{\mathrm{MI}}$ and Multi MAVEN, we use $n_{\mathrm{seed}} = 8$ and $|z| = 8$.

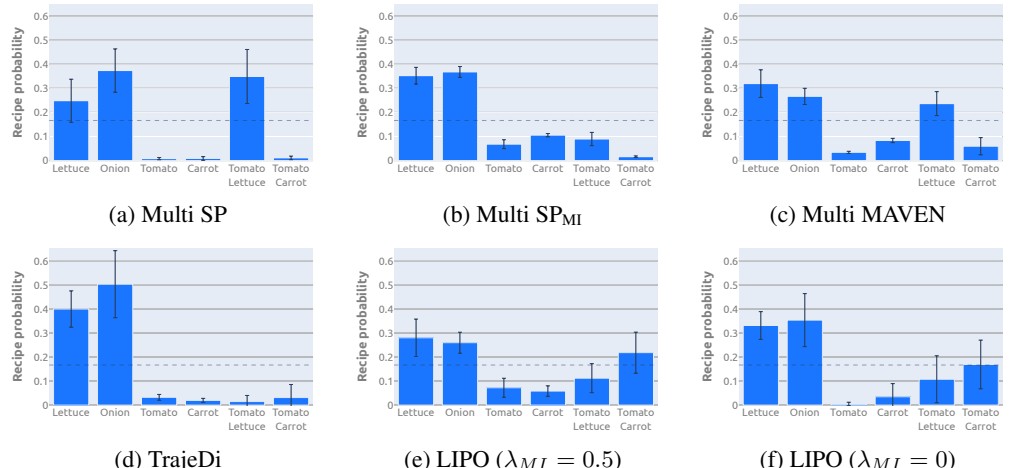

Figure 8: Recipe distributions of generated populations compared to the uniform distribution (dashed line). We provide a reference for the uniform distribution as it has the highest entropy.

Table 1: The means and standard errors entropy of approximated population recipe distributions.

| Multi SP | TrajeDi | Multi SP$_{MI}$ | Multi MAVEN | LIPO ($\lambda_{MI} = 0.5$) | LIPO ($\lambda_{MI} = 0$) |
|---|---|---|---|---|---|
| $1.16 \pm 0.03$ | $0.98 \pm 0.17$ | $1.43 \pm 0.05$ | $1.53 \pm 0.07$ | $\mathbf{1.58} \pm 0.07$ | $1.26 \pm 0.17$ |

**Results:** Quantitatively, Tab. 1 shows that LIPO has the highest population recipe distribution entropy, averaged over five random seeds. This result indicates that LIPO populations use all recipes more uniformly than the baseline populations, even though some recipes take longer to complete or are harder to find by random exploration. The recipe distribution of populations produced by each method can be found in Fig. 8. We find that LIPO populations with $\lambda_{MI} = 0$ still, similar to $\lambda_{MI} = 0.5$, consistently learn to use the hard-to-find Tomato & Carrot Salad recipe. However, the frequencies of Chopped Tomato and Chopped Carrot are lowered (Fig. 8f). This means that there are multiple joint policies that learn to complete the same recipe while being incompatible with each other. We suspect that using $\lambda_{MI} > 0$ alleviates this problem by regularizing each joint policy to represent a policy with broader state-action coverage (e.g., learn multiple ways of completing a recipe), indirectly pushing other joint policies to use different recipes in order to be incompatible. Qualitatively, we can see in App. J.2 that the baselines produce agents with similar recipe frequencies. The resulting populations, thus, contain agents with a similar recipe preference. In contrast, LIPO produces agents with distinct recipe frequencies, collectively making the population more diverse than the baselines. We also visualize the behaviors learned by LIPO in App. J.3.

## 4.6 TRAINING GENERALIST AGENTS WITH GENERATED POPULATIONS

In addition to evaluating the diversity of agents in Overcooked, we quantify the usefulness of the produced agents by using them as training partners of a generalist agent and test the agent with held-out populations. Intuitively, more diverse training partners would enable the agent to generalize and coordinate with unseen agents better. Additionally, we include a population of six specialized SP policies where each policy is trained to complete a specific recipe by adjusting the reward function. The specialist population is created for evaluating the agent when the partner has a strong preference.

Fig. 9a shows that all generalist agents perform similarly when tested with held-out baseline populations. However, the agents trained with a baseline population perform poorly when matched with held-out LIPO and specialist populations. In contrast, those trained with a LIPO population perform better in both situations. Specifically, they have a significantly higher success rate when paired with the specialist with a strong preference for Tomato & Carrot Salad as shown in Fig. 9b. We attribute the success rate difference to the fact that this recipe has a lower completion probability in all except LIPO populations. As a result, generalist agents trained with a LIPO population perform better in terms of the overall success rate when tested with specialist agents. Overall, training with a LIPO population helps the generalist agents to better coordinate with more partner types as indicated by the harmonic means.

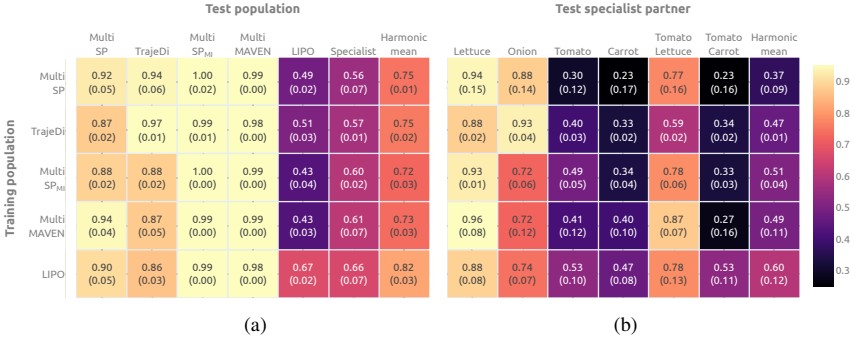

Figure 9: The mean success rates and standard errors (in parentheses) of trained generalist agents when matched with unseen test agents. Each generalist agent is trained with only one population and evaluated with all test partners, each with 300 episodes. The result in each row is averaged over five generalist agents trained with an independently generated population from the corresponding method. A held-out population from each method is used as a test population. The right-most column shows the harmonic mean of the success rate of all held-out populations (a) and specialist agents (b).

## 5 RELATED WORK

Learning a collection of diverse agents has been utilized in various contexts (Parker-Holder et al., 2020; Sun et al., 2020; Zahavy et al., 2021; Zhou et al., 2021). In the cooperative domain, Canaan et al. (2019; 2020) use the Quality Diversity (QD) algorithm (Mouret & Clune, 2015; Pugh et al., 2016) to produce a population of behaviorally diverse agents. QD, however, requires domain knowledge to encode different types of behaviors. For example, in CMG, the algorithm requires the mapping between actions and corresponding solutions. In Overcooked, it needs to know all possible recipes beforehand. Without such a domain knowledge, it would be difficult to use QD to produce a diverse population. TrajeDi (Lupu et al., 2021) produces a diverse population of agents based on the trajectory distribution. Finally, MEP (Zhao et al., 2021) trains a population of agents with an auxiliary reward based on population entropy. Like TrajeDi and MEP, ours does not require domain-specific knowledge. However, to promote behavioral diversity, LIPO utilizes the expected returns of different policy pairs as opposed to state-action information.

The idea of diversifying the empirical return has been explored in the context of finding diverse solutions in non-transitive competitive games (Liu et al., 2021; Balduzzi et al., 2019; Perez-Nieves et al., 2021). In particular, Liu et al. (2021) share some similar ideas with our work. They propose to use the expected returns, when encountering different opponents, and state-action information to promote diversity of agents. A concurrent work by Rahman et al. (2022) applies a similar idea of diversifying the expected joint return to generate diverse partners in cooperative settings. LIPO can be thought of as a special case designed specifically for cooperative environments (see App. E).

MI objectives have been used in RL to learn diverse behaviors (Eysenbach et al., 2018; Sharma et al., 2019; Kumar et al., 2020; Osa et al., 2022). In cooperative MARL, MAVEN (Mahajan et al., 2019) optimizes both RL and MI objectives to encourage the agents to explore in a committed manner and discover diverse solutions. Also, Any-play (Lucas & Allen, 2022) uses a similar objective to produce training partners with many solutions for a generalist agent. In contrast, our approach uses the MI objective to regularize each policy to learn local variations of each solution.

## 6 CONCLUSION

We propose LIPO, a simple and generic method that can create a population of diverse agents in cooperative multi-agent environments. Unlike previous work that uses state-action information from joint trajectories, LIPO utilizes the concept of policy compatibility to create diverse policies. This alternative view of quantifying diversity makes LIPO more robust to state and action spaces. Also, LIPO uses the MI objective to learn local variations of each solution. Empirically, LIPO consistently produces more diverse populations than the baselines across a variety of three multi-goal environments. Finally, in multi-recipe Overcooked, LIPO produces populations of diverse partners that help the generalist agents to generalize to unseen agents better. We include further discussions and limitations of LIPO in App. F and G.

ACKNOWLEDGEMENT

This work is partially supported by King Mongkut's Institute of Technology Ladkrabang [2566-02-06-002]. We thank Natchaya Sricom for drawing Fig. 1 and 7. We thank Supasorn Suwajanakorn, Sucha Supittayapornpong and Maytus Piriyajitakonkij for their suggestions on early draft versions. We also thank anonymous reviewers for their constructive feedbacks.

REPRODUCIBILITY STATEMENT

We have include additional information to reproduce the experimental results in the supplementary text:

- Environment details (App. C)
- Pseudocode and implementation details (App. D)
- Hyperparameters used in all experiments (App. H and I)

The source code is available at https://github.com/51616/marl-lipo.

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

# A    PROOF FOR THEOREM 3.1

We prove the relationship of similar policies and compatible policies in Theorem 3.1 under the following assumptions.

**Assumption A.1** (All joint trajectories are supported by $\pi_B$). $P(\tau|\pi_B^i, \pi_B^j) > 0; \forall \tau \in \mathcal{T}$

**Assumption A.2** (Shared $\epsilon$). A common $0 \leq \epsilon \leq 1$ is used for Def. 2.1 and 2.2

**Assumption A.3** (Positive return). $G(\tau) > 0; \forall \tau \in \mathcal{T}$

**Theorem.** *If $\pi_A^i$ is similar to $\pi_B^i$, then $\pi_A^i$ is compatible with $\pi_B$.*

*Proof.* Let $r(\tau) = \frac{P(\tau|\pi_A^i, \pi_B^j)}{P(\tau|\pi_B^i, \pi_B^j)}$. Because $\pi_A^i$ is similar to $\pi_B^i$, we know that $1 - \epsilon \leq r(\tau) \leq 1 + \epsilon; \forall \tau$ from Def. 2.1. Consequently, we can use importance sampling to write $J(\pi_A^i, \pi_B^j)$ in relation to $J(\pi_B^i, \pi_B^j)$:

$$
\begin{aligned}
J(\pi_A^i, \pi_B^j) &= \mathbb{E}_{\tau \sim \rho(\pi_A^i, \pi_B^j)} G(\tau) \\
&= \int_\tau P(\tau|\pi_A^i, \pi_B^j) G(\tau) \\
&= \int_\tau \frac{P(\tau|\pi_B^i, \pi_B^j)}{P(\tau|\pi_B^i, \pi_B^j)} P(\tau|\pi_A^i, \pi_B^j) G(\tau) \\
&= \int_\tau r(\tau) P(\tau|\pi_B^i, \pi_B^j) G(\tau) \\
&= \mathbb{E}_{\tau \sim \rho(\pi_B^i, \pi_B^j)} r(\tau) G(\tau)
\end{aligned}
$$

Because $1 - \epsilon \leq r(\tau) \leq 1 + \epsilon$ and $G(\tau) > 0; \forall \tau \in \mathcal{T}$, the expected return $J(\pi_A^i, \pi_B^j)$ has the following upper and lower bounds:

$$
\begin{aligned}
(1 - \epsilon) \mathbb{E}_{\tau \sim \rho(\pi_B^i, \pi_B^j)} G(\tau) \leq &J(\pi_A^i, \pi_B^j) \leq (1 + \epsilon) \mathbb{E}_{\tau \sim \rho(\pi_B^i, \pi_B^j)} G(\tau) \\
(1 - \epsilon) J_{\text{SP}}(\pi_B) \leq &J(\pi_A^i, \pi_B^j) \leq (1 + \epsilon) J_{\text{SP}}(\pi_B)
\end{aligned}
$$

This means that $\pi_A^i$ is compatible with $\pi_B$ (Def. 2.2).

$\therefore$ If $\pi_A^i$ is similar to $\pi_B^i$, then $\pi_A^i$ is compatible with $\pi_B$.    $\square$

**Remark:**    Assumption A.3 can be satisfied by offsetting the joint return of all trajectories such that $\min_\tau G(\tau) > 0$. However, since we use MAPPO as the base algorithm, the expected return is subtracted by a baseline to compute the advantage during the policy update, which removes the effect of the offset. In practice, even under environments that do not satisfy $G(\tau) > 0$, LIPO can still discover diverse solutions effectively, as shown in the experiments.

# B    DERIVATION OF THE LOWER BOUND OF THE MI OBJECTIVE

We provide derivation of Eq. 7 here. Let $p(z^i|\{o^i, a^i\})$ be the true posterior of $z^i$ and $q_\phi$ be the approximation of $p$ parameterized by $\phi$. The lower bound of $I(\{o^i, a^i\}|z^i)$ can be derived as follows:

$$
\begin{aligned}
I(\{o^i, a^i\}; z^i) &= H(z^i) - H(z^i|\{o^i, a^i\}) \\
&= H(z^i) + \mathbb{E}_{z^i, (o^i, a^i)}[\log p(z^i|\{o^i, a^i\})] \\
&= H(z^i) + \mathbb{E}_{z^i, (o^i, a^i)}[\log p(z^i|\{o^i, a^i\}) - \log q_\phi(z^i|\{o^i, a^i\}) + \log q_\phi(z^i|\{o^i, a^i\})] \\
&= H(z^i) + \mathbb{E}_{z^i, (o^i, a^i)}[\log q_\phi(z^i|\{o^i, a^i\})] + \text{KL}(p(z^i|\{o^i, z^i\})||q_\phi(z^i|\{o^i, z^i\})) \\
I(\{o^i, a^i\}; z^i) &\geq H(z^i) + \mathbb{E}_{z^i, (o^i, a^i)}[\log q_{\phi_A}(z^i|\{o^i, a^i\})]
\end{aligned}
$$

## C  ADDITIONAL ENVIRONMENT DETAILS

### C.1  ONE-STEP COOPERATIVE MATRIX GAME

A game of CMG is defined by a tuple $(M, \{k_m\}, \{r_m\})$, where $M$ is the number of solutions. For $m \in \{1, ..., M\}$, $k_m$ is the number of compatible actions and $r_m$ is the reward of a solution $m$. The game is stateless and terminate immediately after both players simultaneously choose an action. By choosing the same solution, both players get a reward $r_m$ associated with the chosen solution. This means that the solutions are not equally optimal if the values in $\{r_m\}$ are not identical. Similarly, if the values in $\{k_m\}$ are not identical then the solutions are not equally likely to be chosen by a uniform joint policy. We consider two setups of CMG: sub-optimal (CMG-S) and hard-to-find (CMG-H).

For CMG-S, we set $(M = 32, k_m = 8, r_m = 0.5 * (1 + \frac{m-1}{M-1}))$, which causes each solution to have a different reward, ranging from 0.5 to 1. There are 32 solutions, each with 8 compatible actions. There are $32 \times 8 = 256$ possible actions for each player.

For CMG-H, we use $(M = 32, k_m = m, r_m = 1)$, which makes solutions with a smaller number of compatible actions harder to be found by random exploration. There are 32 solutions, each with different number of compatible actions, ranging from 1 to 32. The number of available actions for each player is $\sum_{m=1}^{m=32} m = 528$.

### C.2  POINT MASS RENDEZVOUS (PMR)

PMR is based on the the Multi-Agent Particle Environment (Lowe et al., 2017; Terry et al., 2020). The observation of each agent includes absolute position, current velocity, and the relative distance to the landmarks and the other agent. These features are concatenated as a 1-D vector of length 14. The possible actions are: `no op`, `move`, $\{$`up`, `down`, `left`, `right`$\}$. In PMR-C, the start positions of the agents are $\{(0.3,0), (-0.3,0)\}$ and the landmarks positions are $\{(1.59, 1.59), (1.59, -1.59), (-1.59, 1.59), (-1.59, -1.59)\}$. For PMR-L, the start and the landmark positions are $\{(1,0),(0,1)\}$ and $\{(0,2.25),(0,0.75),(0,-0.75),(0,-2.25)\}$. An episode will be terminated after 50 timesteps. The agents are incentivized to go to the same landmark and stay close together with the reward function

$$r_t = 1 - d(p^i, c) - \min_{l \in L} d(l, c),$$

where $d(\cdot, \cdot)$ is the euclidean distance between two points, $p^i$ is the 2-d coordinate of agent $i$, $c$ is the average coordinate of all agents, and $L$ is the set of all landmarks.

### C.3  MULTI-RECIPE OVERCOOKED

We implement a multi-recipe of the game based on the work of Wu et al. (2021). In this version of Overcooked, there are four ingredients: lettuce, onion, tomato, and carrot. The ingredients are randomly placed at pre-defined positions in the layout. Particularly, the lettuce and the onion are randomly placed on the left or the middle counter. The tomato and the carrot are randomly placed on the right or the middle counter. These ingredients can be composed into different recipes making each ingredient unique: four recipes (`LettuceSalad`, `TomatoSalad`, `ChoppedCarrot`, `ChoppedOnion`) require only a single ingredient, while the other two (`TomatoLettuceSalad`, `TomatoCarrotSalad`) require two ingredients. The ingredients have to be chopped at the chopping station before placing on the plate. After the required ingredients are put on the plate, they must be delivered to the delivery station.

Both players have the same egocentric observation and action spaces. The observation is a set of hand-crafted features that represent a local view of the environment. Specifically, we use the following features: absolute position and facing direction, relative distance to the objects and the other agent, state of the ingredients, four booleans indicating if the agent is next to a counter in four cardinal positions, currently held items, the state of the held foods, and the type and state of the items in front of the agent. These features are concatenated as a 1-D vector of length 54. At every timestep, each player has to choose one of the six possible actions: `no op`, `move` $\{$`up`, `down`, `left`, `right`$\}$, and `interact`.

An episode lasts at most 200 timesteps and terminates immediately after a successful delivery. An episode without delivery is considered unsuccessful. We incentivize the agents to interact with the

objects and deliver as fast as possible with the following reward function:

$$r_t = r_{\text{interact}} + r_{\text{progress}} + r_{\text{complete}} - p,$$

here $r_{\text{interact}}$ is a shaped reward given when an agent interacts with an object for the first time in an episode, $r_{\text{progress}}$ is given when the players progress toward a recipe completion (i.e., chopping required ingredients or putting chopped ingredients on the plate), $r_{\text{complete}}$ is given upon successful delivery, and $p$ is a penalty. We use $r_{\text{interact}} = 0.5$, $r_{\text{progress}} = 1.0$, $r_{\text{complete}} = 10$, and $p = 0.1$. We note that recipes with more than one ingredient will give only slightly higher rewards ($r_{\text{interact}} + r_{\text{progress}}$) but are significantly harder to be discovered by random exploration than those with one ingredient.

**Additional experimental details:** For specialist agents, the rewards are given when interacting, progressing, or completing a specific recipe. For the held-out populations, we remove incompetent policies with the expected return of less than zero from the test populations created by TrajeDi and LIPO. We do not remove those in the training populations. We do this because testing with an incompetent policy does not give any meaningful information, as almost all episodes will be unsuccessful.

## D  IMPLEMENTATION DETAILS

---

**Algorithm 1:** Training process of LIPO (on-policy)

This pseudocode is based on self-play. Blue text is related to the MI objective. LIPO specific code is highlighted in green.

**Input:** A Population $\mathcal{P} = \{\pi_A \,|\, 1 \leq A \leq N\}$, the number of XP pairs used to approximate $\tilde{J}_{xp}(\pi_A, \mathcal{P})$ ($n_{xp}$), the number of players in an episode ($m$), and the number of SP and XP episodes per iteration ($E_{\text{SP}}$ and $E_{\text{XP}}$).

**while** *not done* **do**

    **for** $A \in \{1, ..., N\}$ **do**

        $\mathcal{B}^{sp} \leftarrow \text{GetEpisodeRollouts}(\pi_A, \pi_A, E_{\text{SP}}, m)$

        Compute $J_{\text{SP}}(\pi_A)$ using $\mathcal{B}^{sp}$

        $\mathcal{B}^{xp} \leftarrow \text{GetCrossPlayRollouts}(\pi_A, \mathcal{P}, n_{xp}, E_{\text{XP}}, m)$

        Compute $\tilde{J}_{\text{XP}}(\pi_A, \mathcal{P})$ using $\mathcal{B}^{xp}$ (Eq. 4)

        Compute $L_{\text{MI}}$ using $\mathcal{B}^{sp}$ and $\mathcal{B}^{xp}$ (Eq. 9)

        $\theta_A \leftarrow \theta_A - \nabla_{\theta_A}[-J_{\text{SP}} + \lambda_{\text{XP}}\tilde{J}_{\text{XP}} + \lambda_{\text{MI}}L_{\text{MI}}]$

        $\phi_A \leftarrow \phi_A - \lambda_{\text{MI}}\nabla_{\phi_A}L_{\text{MI}}$

---

**Algorithm 2:** Rollout collection functions

**Function** `GetEpisodeRollouts`$(\pi_A, \pi_B, E)$**:**

    $\mathcal{B} \leftarrow \{\}$

    **for** $i \in \{1, ..., m\}$ **do**

        **for** $episode \in \{1, .., \frac{E}{m}\}$ **do**

            $z^1, ..., z^m \sim p(z^1, ..., z^m)$

            $z^j \leftarrow \{z^k\}_{k \neq i}^m$

            $\pi^j(\cdot|\cdot, z^j) = \Pi_{k \neq i}\pi^k(\cdot|\cdot, z^k)$

            $\tau \sim \rho(\pi_A^i(\cdot|\cdot, z^i), \pi_B^j(\cdot|\cdot, z^j))$

            $\mathcal{B} \leftarrow \mathcal{B} \cup \{\tau\}$

    **return** $\mathcal{B}$

**Function** `GetCrossPlayRollouts`$(\pi_A, \mathcal{P}, n_{xp}, E_{\text{XP}})$**:**

    $\mathcal{B}^{xp} \leftarrow \{\}$

    $\mathcal{P}'_{-A} \leftarrow \text{SampleWithoutReplacement}(\mathcal{P}_{-A}, n_{xp})$

    **for** $\pi_B \in \mathcal{P}'_{-A}$ **do**

        $\mathcal{B} \leftarrow \text{GetEpisodeRollouts}(\pi_A, \pi_B, \frac{E_{\text{XP}}}{|\mathcal{P}'_{-A}|})$

        $\mathcal{B}^{xp} \leftarrow \mathcal{B}^{xp} \cup \mathcal{B}$

    **return** $\mathcal{B}^{xp}$

---

The pseudocode for LIPO is shown in Algorithm 1 and 2. If there are more than two players ($m > 2$), $\pi^j$ would represent the joint policy of all players except player $i$, $\pi^j(a_t^j|\tau_t^j) = \Pi_{k \neq i}\pi^k(a_t^k|\tau_t^k)$. We note that scaling LIPO to more than two players does not increase the training time. It is the same as the two-player setting as long as the numbers of SP and XP episodes are the same. In practice,

Table 2: Hyperparameters used by the MAPPO algorithm.

| Hyperparameters | Value |
|---|---|
| Learning rate | 0.003 (CMG and PMR) 
 0.005 (Overcooked) |
| Batch size | 100 (CMG) 
 2,500 (PMR) 
 10,000 (Overcooked) |
| Epochs | 10 (CMG and PMR) 
 15 (Overcooked) |
| Number of mini-batches | 2 (CMG and PMR) 
 5 (Overcooked) |
| Entropy coefficient | 0.0 (CMG) 
 0.03 (PMR and Overcooked) |
| Discount factor ($\gamma$) | 0.99 |
| GAE lambda | 0.95 |
| Value loss coefficient | 0.5 |
| PPO clipping parameter | 0.3 |
| Gradient clipping | 0.5 |
| Adam epsilon | 1e-5 |

however, more XP episodes might be needed to better estimate $\tilde{J}_{\text{XP}}$. We use the parameter sharing technique for better sample efficiency and faster convergence (Tan, 1993; Foerster et al., 2018; Rashid et al., 2018). Assuming that a policy $\pi_A^i$ is a neural network parameterized by $\theta_A^i$, this means that for a joint policy $(\pi_A^1, \pi_A^2)$, we have $\theta_A^1 = \theta_A^2$. Still, $\pi_A^1$ and $\pi_A^2$ can behave differently as they observe different parts of the environment and have a different player indicator concatenated with their local observations.

All methods are implemented on top of MAPPO except MAVEN. The critic, policy, and discriminator are feed-forward neural networks with two hidden layers, each having 64 units. For a fair comparison, we use the same or more environment steps in the policy update of the baselines compared to LIPO. Common hyperparameters of methods based on MAPPO are shown in Table. 2.

### D.1 MAPPO

MAPPO is the base MARL algorithm for all baselines except MAVEN. The policy parameters are shared among all policies. The critic takes a state of the environment and outputs an expected return of a given global state. The global state is provided by the environment and only used during training. For the complete training objectives of MAPPO, we refer the reader to Appendix A of Yu et al. (2021).

### D.2 MULTI SP

A simple but effective way to produce diverse agents by training multiple SP agents with different neural network initializations and random seeds. Specifically, each run produces a joint policy $\pi_A$ that maximizes $J_{\text{SP}}(\pi_A)$ using MAPPO.

### D.3 SP$_{\text{MI}}$

A single run of SP agent trained with added MI objective $I(z|o^i, a^i)$. SP$_{\text{MI}}$ uses a shared $z$ for both policies and considers each $z$ as a different joint policy. We train SP$_{\text{MI}}$ using the same training procedure as LIPO by setting $N = 1$, $\lambda_{\text{XP}} = 0$ and $z^1 = z^2$. The discriminator takes a local

observation $o^i$ and action distribution $\pi^i(\cdot|o^i)$ as inputs and outputs the discrete probability of the latent variable. The latent variable of all policies is shared during an episode.

## D.4 MAVEN

MAVEN (Mahajan et al., 2019) is explicitly designed for learning diverse solutions in cooperative multi-agent environments. A joint policy is represented as $\pi(\cdot|\tau, z)$, and each mode of behavior is represented by the latent variable $z$. Similar to $\text{SP}_{\text{MI}}$, MAVEN uses a shared $z$ for all policies. We use the same network architecture presented in Mahajan et al. (2019) with recurrent neural networks. However, we do not use the hierarchical policy but sample $z$ from the uniform distribution. The latent variable of all policies is shared.

## D.5 MULTI $\text{SP}_{\text{MI}}$ AND MULTI MAVEN

A population containing joint policies from multiple runs of $\text{SP}_{\text{MI}}$ and MAVEN. Like Multi SP, each run has different neural network initializations and random seeds. The population size is $|\mathcal{P}| = n_{\text{seed}}|z|$, where $n_{seed}$ is the number of runs. Notably, this baseline uses the training data differently from the base algorithms. Instead of training a long single run, this approach allows the policy to "restart" by using different initialization of neural networks. For example, training a single run with $|z| = N, n_{\text{seed}} = 1$ might not discover as many solutions as training $n_{\text{seed}}$ runs with $|z| = \frac{N}{n_{\text{seed}}}$ even though the population size is the same. Empirically, we find that multiple shorter runs can find more solutions compared to a single long run of the corresponding algorithm. Thus, we omit the results of the base algorithms in multi-recipe Overcooked.

## D.6 TRAJEDI

TrajedDi produces a population of diverse agents that also maximize the expected return in cooperative environments. The diversity measure of this method is based on the Jensen-Shannon divergence (JSD) between the trajectory distribution of each policy. Different from the original implementation, we remove the best-response (BR) policy from the population. Since the BR policy might work well with only a subset of solutions, removing BR potentially increase the number of variations in the population. Our modified loss is:

$$\mathcal{L} = -[\sum_{A=1}^{N}(J_{\text{SP}}(\pi_A)) + \alpha\text{JSD}_\gamma(\pi_1, ..., \pi_N)], \tag{11}$$

where JSD is the proposed diversity objective of TrajeDi, and $\alpha$ and $\gamma$ are the hyperparameters of TrajeDi.

## D.7 LIPO

LIPO uses the same implementation as $\text{SP}_{\text{MI}}$ except LIPO uses independent latent variable for each policy and $\lambda_{\text{XP}} > 0$. Additionally, LIPO uses extra critics for the XP trajectories. In total, LIPO has an SP critic $V_{sp}^{\pi_A}$ and $N - 1$ XP critics $\{V_{xp}^{\pi_A, \pi_B} \mid \pi_B \in \mathcal{P}_{-A}\}$. In each training iteration, LIPO collects SP and XP trajectories of all policy combinations. MAPPO is used for both maximizing $J_{\text{SP}}$ and minimizing $\tilde{J}_{\text{XP}}$. The critics are trained using SP trajectories and all of XP trajectories, while the policy is trained using SP trajectories and XP trajectories from the XP pair that has the highest joint return, $\max(\mathcal{B}^{xp})$.

## D.8 GENERALIST AGENT

The policy and critic networks of a generalist agent use two 256-unit GRU layers (Cho et al., 2014) followed by a linear layer. The input also includes the reward and action of the previous timestep. We use MAPPO for training a generalist agent. We train both the policy and critic with the batch size of 320,000 timesteps using truncated backpropagation through time (BPTT). The samples are reused for 15 epochs. Each minibatch contains 1,600 sequences with a maximum length of 50 timesteps. We also use learning rate annealing, specifically the generalist agent starting from $0.005$ to $0.003$ with linear scheduling. Other hyperparameters are shared with other methods (Tab. 2). A training

partner for a generalist agent is sampled uniformly from the training population at the beginning of an episode.

## E RELATIONSHIP WITH RAHMAN ET AL. (2022)

Rahman et al. (2022) propose to optimize the self-play returns while maximizing the diversity term $\text{Div}(C)$, where C is a $N \times N$ cross-play payoff matrix. Specifically, they propose to learn diverse policies via the following objective:

$$\max_{C} \text{Tr}(C) + \text{Div}(C) \tag{12}$$

They propose to use $\text{Div}(C) = \text{Det}(\kappa(C))$ where $\kappa(C)$ is an $N \times N$ matrix with $\kappa_{i,j}(C)$ being similarity between policy $\pi_i$ and $\pi_j$. The radial basis function (RBF) kernel of the empirical returns is used to measure the similarity between two policies:

$$\kappa_{i,j}(C) = \exp(-\frac{||C_{i,\cdot} - C_{j,\cdot}||^2}{\sigma^2}), \tag{13}$$

where $C_{i,\cdot}$ is row $i^{th}$ of C. In other words, $C_{i,\cdot}$ is the vector containing the empirical return of $\pi_i$ when matched with other policies in the population. Intuitively, this objective diversifies the policies via the expected returns similar to LIPO. Using the same notation of the cross-play matrix, we can write the objective of training a LIPO population as:

$$\max_{C} \text{Tr}(C) - \lambda_{\text{XP}} \sum_{i} \max_{\substack{1 \leq j \leq N \\ i \neq j}} C_{i,j} \tag{14}$$

This objective wants the diagonal ($J_{\text{SP}}$) to be maximized and the off-diagonal entries of the cross-play matrix ($\tilde{J}_{\text{XP}}$) to be minimized. This is a special case of Eq. 12 where $\text{Div}(C)$ is based on policy compatibility.

## F DISCUSSIONS

Agents trained with LIPO are incentivized to act adversarially toward agents that behave differently from itself. This behavior might not be desirable for certain downstream tasks. For example, agents produced by LIPO might not be suitable for interacting with humans as they would refuse to conform with the user. However, as shown in Sec. 4.6, training a generalist agent with these agents would have the opposite effect: the generalist agent would try to comply with the current partner's preference.

LIPO produces a population of near-optimal solutions, a generalist agent trained with a LIPO population might not coordinate well with significantly sub-optimal agents. In a prior work, Strouse et al. (2021) show that augmenting the training population with past checkpoints (FCP) helps the trained generalist agent to effectively coordinate with sub-optimal agents. Since LIPO and FCP are orthogonal, populations created by LIPO can be also augmented in the same way as FCP.

Previous works find incompatible policies to be undesirable since they are generally results of coordinated symmetry breaking (Bard et al., 2020; Hu et al., 2020; 2021); these policies perform poorly when interacting with unseen partners. However, we show that learning incompatible policies can be useful for generating behaviorally diverse agents in various scenarios. The produced agents can then be used as training partners for a generalist agent. Using LIPO in environments where many solutions are equivalent may produce such undesirable symmetry-breaking conventions. We believe that LIPO can be combined with other techniques, e.g., *other-play* (Hu et al., 2020) and *equivariant coordinator (Muglich et al., 2022)*, to avoid learning arbitrary symmetry-breaking. We leave the study of the combination of LIPO and these techniques for future work. A concurrent work by Cui et al. (2023) proposes an extension of LIPO by combining insights from *off-belief learning* (Hu et al., 2021) to avoid "sabotaging" behavior of LIPO agents.

## G LIMITATIONS

LIPO requires an additional hyperparameter $\lambda_{\text{XP}}$. If $\lambda_{\text{XP}}$ is too big, it is possible that the main RL objective, $J_{\text{SP}}$, would be interfered which will result in an incompetent joint policy (Sec. 4.2). An

adaptive mechanism that selects a suitable value for $\lambda_{\text{XP}}$ at different stages of training could help increase training stability.

Although LIPO can be fully parallelized, it requires more computation than the baselines to get an accurate approximation of $\tilde{J}_{\text{XP}}$, which makes it harder to scale up to bigger population sizes. Instead of collecting all policy pairs, sampling a portion of policy pairs to approximate $\tilde{J}_{\text{XP}}$ could reduce computation cost and training time at a potential cost of diversity (Sec. 4.3). Instead of using a uniform sampling, a mechanism that selects the best pair to sample (e.g., bandit algorithm) might help mitigate the diversity loss from using lower $n_{xp}$.

In this work, we only investigate the effectiveness of LIPO under a new set of cooperative environments with multiple discrete solutions. Investigating LIPO in other well-known environments where the solution space might be continuous, e.g., continuous control (Peng et al., 2021), or environments with only a single goal, e.g., SMAC (Samvelyan et al., 2019), might yield interesting results and pose different challenges. We leave this further investigation as our future work.

## H  HYPERPARAMETERS (CMG AND PMR)

We provide the searched values of each method in Tab. 3, 4, 5, 6, 7 and 8. The hyperparameters are searched individually for each population size. We use three random seeds for each set of hyperparameters. We do not use any validation method. Instead, we present the results using the best parameters in the main paper. For LIPO, we set $\lambda_{\text{MI}}$ as 0.0 and 0.5 in CMG and PMR, respectively.

Table 3: The values of $\lambda_{\text{MI}}$ used by SP$_{\text{MI}}$ in all environments. The best values are shown in bold.

| Population size | Hyperparameters | Values (CMG-S) | Values (CMG-H) | Values (PMR-C) | Values (PMR-L) |
|---|---|---|---|---|---|
| 1 | | - | - | [0.5,1,5,**10**] | [0.5,**1**,5,10] |
| 2 | | - | - | [0.5,1,5,**10**] | [0.5,1,**5**,10] |
| 4 | | - | - | [0.5,**1**,5,10] | [0.5,1,**5**,10] |
| 8 | $\lambda_{\text{MI}}$ | [1,5,**10**,50] | [1,5,10,**50**] | [0.5,**1**,5,10] | [0.5,1,**5**,10] |
| 16 | | [1,5,10,**50**] | [1,5,10,**50**] | - | - |
| 32 | | [1,5,10,**50**] | [1,5,10,**50**] | - | - |
| 64 | | [1,5,10,**50**] | [1,5,10,**50**] | - | - |

Table 4: The values of $\lambda_{\text{MI}}$ used by MAVEN in all environments. The best values are shown in bold.

| Population size | Hyperparameters | Values (CMG-S) | Values (CMG-H) | Values (PMR-C) | Values (PMR-L) |
|---|---|---|---|---|---|
| 1 | | - | - | [1,5,**10**,50] | [**1**,5,10,50] |
| 2 | | - | - | [1,5,**10**,50] | [1,5,**10**,50] |
| 4 | | - | - | [**1**,5,10,50] | [1,**5**,10,50] |
| 8 | $\lambda_{\text{MI}}$ | [**1**,5,10,50] | [1,**5**,10,50] | [**1**,5,10,50] | [**1**,5,10,50] |
| 16 | | [1,**5**,10,50] | [1,5,**10**,50] | - | - |
| 32 | | [1,**5**,10,50] | [1,5,**10**,50] | - | - |
| 64 | | [1,5,**10**,50] | [1,5,10,**50**] | - | - |

Table 5: The values of $\lambda_{\text{MI}}$ used by Multi SP$_{\text{MI}}$ in all environments. The best values are shown in bold.

| Population size | Hyperparameters | Values (CMG-S) | Values (CMG-H) | Values (PMR-C) | Values (PMR-L) |
|---|---|---|---|---|---|
| 1 | | - | - | - | - |
| 2 | | - | - | - | - |
| 4 | | - | - | - | - |
| 8 | $\lambda_{\text{MI}}$ | [1,5,10,**50**] | [1,5,**10**,50] | [**1**,5,10,50] | [**1**,5,10,50] |
| 16 | | [1,5,10,**50**] | [1,5,10,**50**] | [1,**5**,10,50] | [1,**5**,10,50] |
| 32 | | [1,5,10,**50**] | [1,5,10,**50**] | [1,**5**,10,50] | [1,**5**,10,50] |
| 64 | | [1,5,10,**50**] | [1,5,10,**50**] | - | - |

Table 6: The values of $\lambda_{\mathrm{MI}}$ used by Multi MAVEN in all environments. The best values are shown in bold.

| Population size | Hyperparameters | Values (CMG-S) | Values (CMG-H) | Values (PMR-C) | Values (PMR-L) |
|---|---|---|---|---|---|
| 1 | | - | - | - | - |
| 2 | | - | - | - | - |
| 4 | | - | | - | - |
| 8 | $\lambda_{\mathrm{MI}}$ | [**1**,5,10,50] | [**1**,5,10,50] | [1,5,**10**,50] | [1,5,**10**,50] |
| 16 | | [**1**,5,10,50] | [**1**,5,10,50] | [1,5,**10**,50] | [1,5,**10**,50] |
| 32 | | [**1**,5,10,50] | [**1**,5,10,50] | [1,5,**10**,50] | [1,**5**,10,50] |
| 64 | | [**1**,5,10,50] | [1,**5**,10,50] | | |

Table 7: The values of $\alpha$ and $\gamma$ used by TrajDi in all environments. The best values are shown in bold.

| Population size | Hyperparameters | Values (CMG-S) | Values (CMG-H) | Values (PMR-C) | Values (PMR-L) |
|---|---|---|---|---|---|
| 1 | | - | - | [1,5,**10**,50] | [1,5,10,**50**] |
| 2 | | - | - | [1,5,10,**50**] | [1,5,10,**50**] |
| 4 | | - | | [**1**,5,10,50] | [1,**5**,10,50] |
| 8 | $\alpha$ | [**0.01**,0.05,0.1,0.2] | [0.01,0.05,**0.1**,0.2] | [1,5,**10**,50] | [1,**5**,10,50] |
| 16 | | [0.01,**0.05**,0.1,0.2] | [0.01,0.05,**0.1**,0.2] | - | - |
| 32 | | [**0.01**,0.05,0.1,0.2] | [0.01,0.05,0.1,**0.2**] | - | - |
| 64 | | [**0.01**,0.05,0.1,0.2] | [0.01,0.05,**0.1**,0.2] | - | - |
| 1 | | - | - | [0, **0.1**, 0.5] | [0, **0.1**, 0.5] |
| 2 | | - | - | [0, **0.1**, 0.5] | [**0**, 0.1, 0.5] |
| 4 | | - | - | [**0**, 0.1, 0.5] | [0, **0.1**, 0.5] |
| 8 | $\gamma$ | 0 | 0 | [0, 0.1, **0.5**] | [0, **0.1**, 0.5] |
| 16 | | 0 | 0 | - | - |
| 32 | | 0 | 0 | - | - |
| 64 | | 0 | 0 | - | - |

Table 8: The values of $\lambda_{\mathrm{XP}}$ used by LIPO in all environments. The best values are shown in bold.

| Population size | Hyperparameters | Values (CMG-S) | Values (CMG-H) | Values (PMR-C) | Values (PMR-L) |
|---|---|---|---|---|---|
| 1 | | - | - | [0.1,0.25,**0.5**,1] | [0.1,0.25,**0.5**,1] |
| 2 | | - | - | [**0.1**,0.25,0.5,1] | [0.1,**0.25**,0.5,1] |
| 4 | | - | - | [0.1,0.25,**0.5**,1] | [0.1,**0.25**,0.5,1] |
| 8 | $\lambda_{\mathrm{XP}}$ | [**0.5**,1] | [0.5,**1**] | [0.1,**0.25**,0.5,1] | [0.1,**0.25**,0.5,1] |
| 16 | | [0.5,**1**] | [0.5,**1**] | - | - |
| 32 | | [0.5,**1**] | [0.5,**1**] | - | - |
| 64 | | [0.5,**1**] | [0.5,**1**] | - | - |

# I   HYPERPARAMETERS (OVERCOOKED)

For each method, we use the parameters that give the highest entropy to generate five populations for Sec. 4.5, and Sec. 4.6. The searched values of each method are:

- TrajeDi: We perform a grid search with following hyperparameters: $\alpha \in \{5, 10\}$ and $\gamma \in \{0, 0.5\}$. We use $\alpha = 5$ and $\gamma = 0.5$ for the results in the paper.

- Multi SP$_{\mathrm{MI}}$: We perform a grid search of $\lambda_{\mathrm{MI}} \in \{5, 10\}$. We use $\lambda_{\mathrm{MI}} = 5$ for the results in the paper.

- Multi MAVEN: We perform a grid search of $\lambda_{\mathrm{MI}} \in \{5, 10\}$. We use $\lambda_{\mathrm{MI}} = 5$ for the results in the paper.

- LIPO: We perform a grid search with following hyperparameters: $\lambda_{\mathrm{XP}} \in \{0.2, 0.3\}$ and $\lambda_{\mathrm{MI}} \in \{0.1, 0.5\}$. We use $\lambda_{\mathrm{XP}} = 0.3$ and $\lambda_{\mathrm{MI}} = 0.5$ for the results in the paper.

# J   ADDITIONAL RESULTS

## J.1   ADDITIONAL ABLATION RESULTS

We provide additional results of numbers of learned solutions with varying N and $\lambda_{\mathrm{XP}}$ (Fig. 10), numbers of competent agents in CMG with varying N and $\lambda_{\mathrm{XP}}$ (Fig. 11), and numbers of learned solutions with varying $n_{xp}$ in CMG (Fig. 12) here. These results are consistent with the analysis presented in Sec. 4.2 and 4.3.

### J.1.1 ADDITIONAL RESULTS WITH VARYING N AND $\lambda_{XP}$

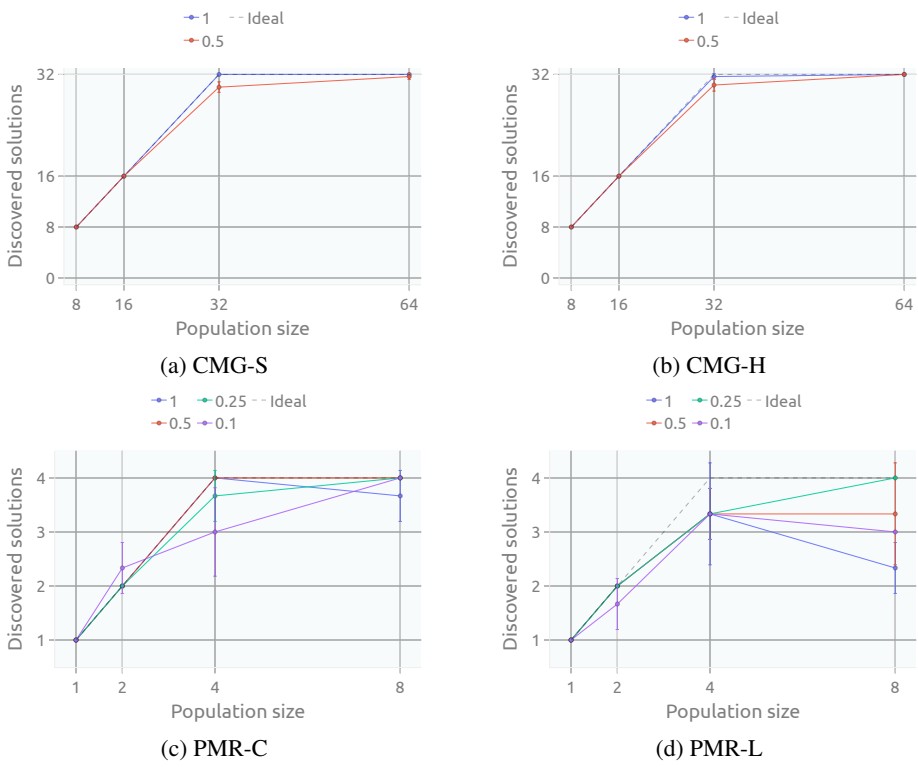

Figure 10: Numbers of learned solutions with different values of $\lambda_{xp}$

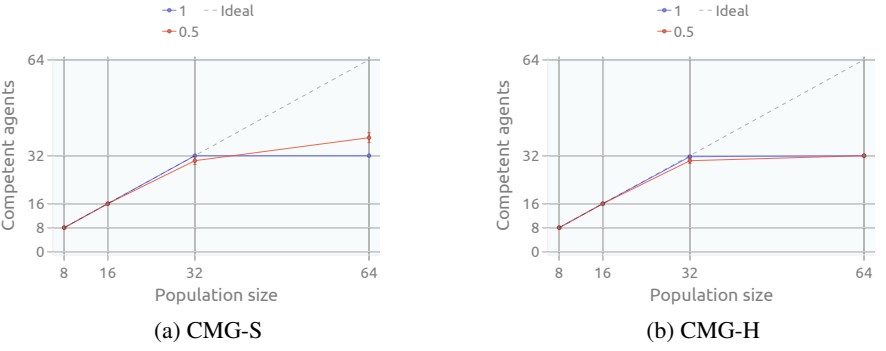

Figure 11: Numbers of competent agents using various combinations of N (x-axis) and $\lambda_{XP}$ (colors) in CMG.

### J.1.2 ADDITIONAL RESULTS WITH VARYING $n_{xp}$

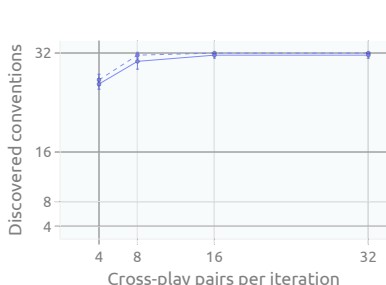

Figure 12: Numbers of learned solutions with varying $n_{xp}$. The result is consistent with the results in PMR provided in the main text: The lower $n_{xp}$ is, the fewer solutions are learned.

## J.2 RADAR PLOTS OF RECIPE FREQUENCIES

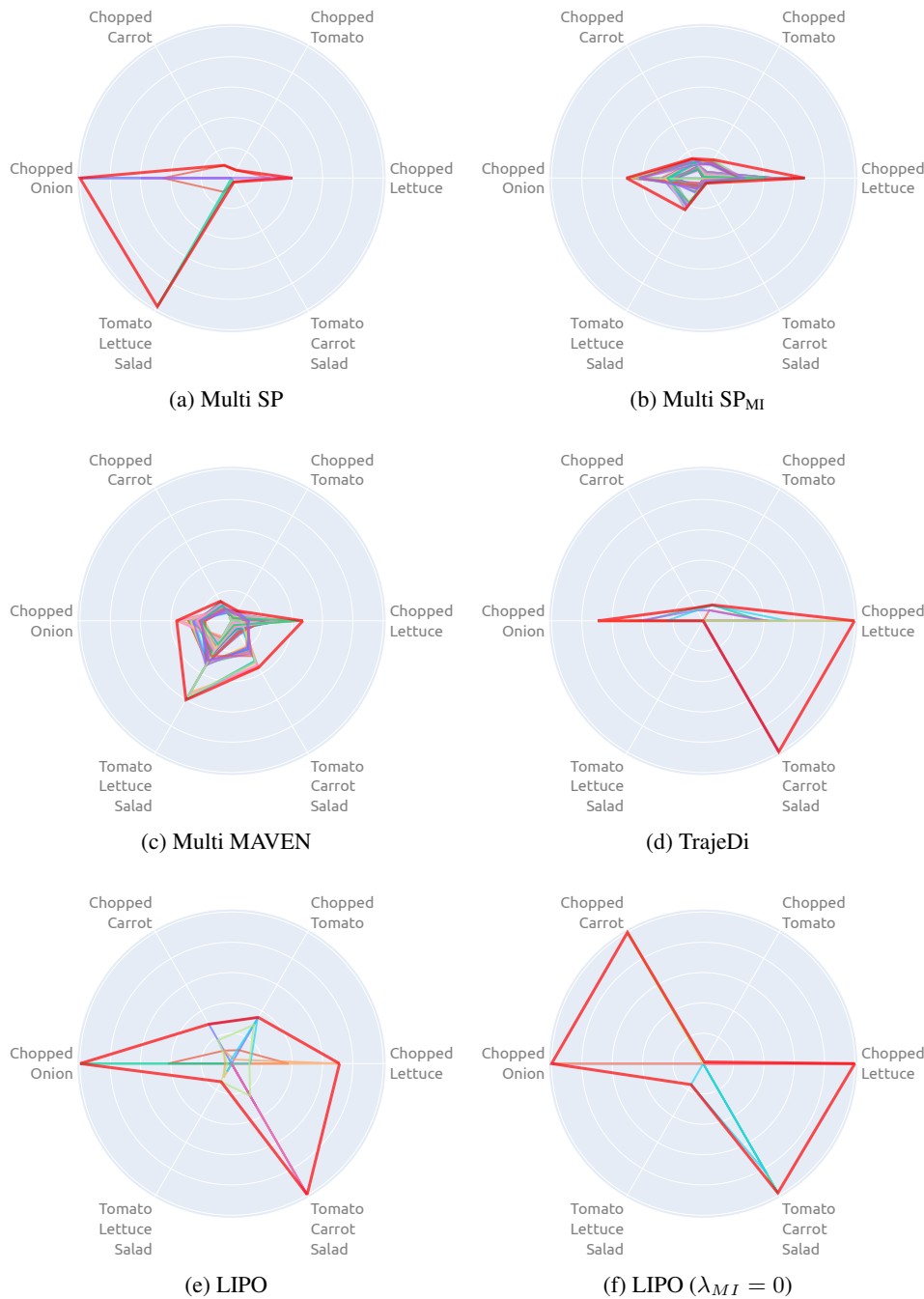

Figure 13: Recipe frequencies of generated populations. Each colored line represents recipe frequencies of an agent in a population. Red thick lines connect the highest frequency of each recipe. The outer ring represents the frequency of 1,000.

Fig. 13 shows recipe frequencies of the population with highest recipe entropy (out of five runs) from each method. The frequencies are calculated based on completed recipe from 1,000 self-play episodes of each agent as described in Sec. 4.5. Qualitatively, we can see that agents in a LIPO

population have more distinct recipe frequencies. That is, agents are different from each other in terms of recipe preference.

### J.3 VISUALIZATION OF BEHAVIORS

We visualize the behaviors joint policies produced by LIPO in PMR and Overcooked at https://sites.google.com/view/iclr-lipo-2023. Here, we show snapshots of four joint policies that have a distinct recipe preference in Overcooked.

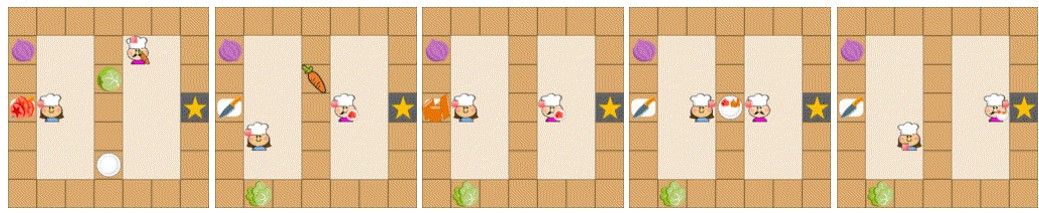

(a) Recipe preference: `Tomato & Carrot Salad`

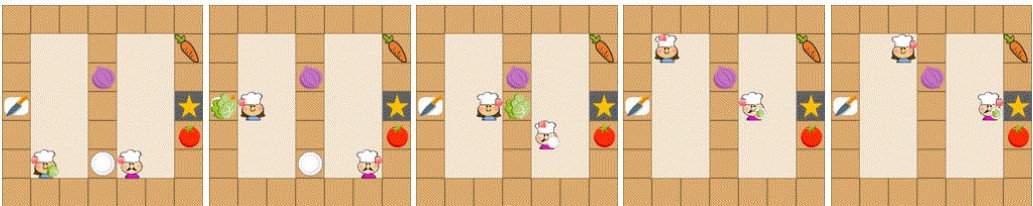

(b) Recipe preference: `Chopped Lettuce`

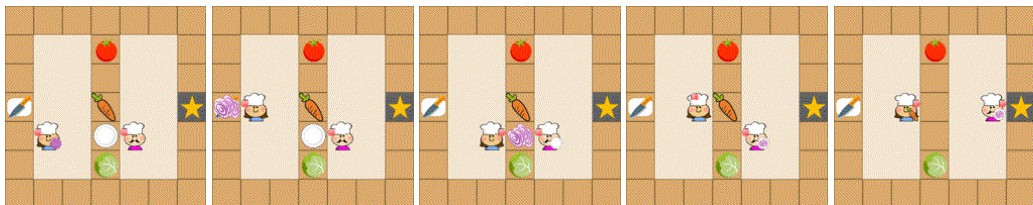

(c) Recipe preference: `Chopped Onion`

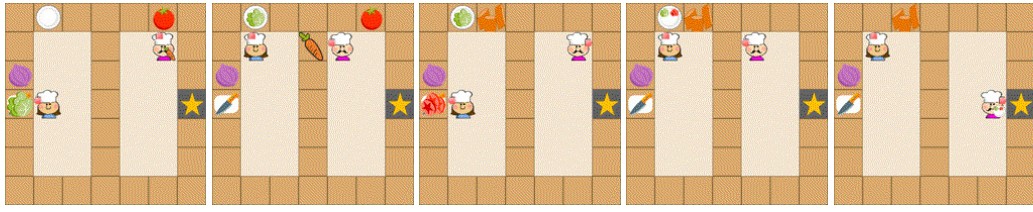

(d) Recipe preference: `Tomato & Lettuce Salad`

Figure 14: Four joint policies from a population of eight joint policies produced by a single run of LIPO. Each row shows snapshots of a joint policy illustrating a distinct recipe preference.

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
