# OpenReview forum: "Generating Diverse Cooperative Agents by Learning Incompatible Policies"
_ICLR.cc/2023/Conference — ICLR 2023 notable top 25%_

### Official Review · Reviewer_obTi · 2022-10-24

**Confidence:** 3
**Correctness:** 4
**Technical Novelty And Significance:** 3
**Empirical Novelty And Significance:** 3
**Recommendation:** 8

**Clarity, Quality, Novelty And Reproducibility:**

The paper is clearly and well-written. It is novel and original, as it uses the concepts of policy compatibility and mutual information in novel manners to create a diverse set of behaviours for cooperative tasks. The work includes a reproducibility statement and is accompanied by the code (however I did not run the code myself).

**Strength And Weaknesses:**

Strength:
- Diverse of benchmarks and baselines
- Comprehensive study of the impact of the parameters
- Ample discussion of results, as well as limitations of the method

Weakness:
- A missing more detailed discussion on scalability, in terms of number of agents, since the current work assumes always 2 agents.

**Summary Of The Paper:**

This work introduces LIPO (Learning Incompatible Policies), a MARL method that uses a policy compatibility-based objective and a mutual information regularizer in order to learn a diverse set of behaviours in cooperative tasks. LIPO is evaluated in a wide set of environments and shows a superior capacity to discover policies against the compared baselines. Additionally, the work also includes a comprehensive study of the parameter values and their effect on the learned behaviours.

**Summary Of The Review:**

All in all, I think this is an interesting and well-presented contribution.

---

> ### Author Response · Authors · 2022-11-13
> **Author response for reviewer obTi**
>
> We thank the reviewer for commenting on the paper as being ***interesting, well presented, well written, and novel.*** We are glad to hear that this manuscript contains ***diverse benchmarks and baselines, a comprehensive study, ample discussion of results, as well as limitations of the method.***
>
> A discussion regarding the scalability to larger numbers of players suggested by the reviewer is added in App. D. Specifically, $\pi_j$ is treated as all other agents, as stated in the footnote of page 2. We updated the pseudocode for applying LIPO to more-than-two-player environments in the revision.

---

### Official Review · Reviewer_PWGa · 2022-10-25

**Confidence:** 4
**Correctness:** 4
**Technical Novelty And Significance:** 3
**Empirical Novelty And Significance:** 3
**Recommendation:** 8

**Clarity, Quality, Novelty And Reproducibility:**

Generally, the paper is very clear and, as mentioned above, the reproducibility of the demonstrated results is high. Since this framework does not specifically require domain knowledge, it can be broadly applicable to multiple unknown settings, where domain knowledge is difficult to obtain. It would be useful to compare LIPO further with MEP (Zhao et al. 2021) to discuss the qualitative/quantitative differences in policies that both obtain.

Line 1 of Appendix A: "proof" should be "prove"

**Strength And Weaknesses:**

Strengths
- While conceptually simple, the proposed framework for generating diverse populations of agents is a powerful idea that could have applicability beyond cooperative domains, e.g., for designing agents that can compete effectively against agents or where optimal strategies are difficult to find.
- The inclusion of the anonymized source code is appreciated, and all of the details in the appendices aid reproducibility.
- The paper is written clearly and includes solid theoretical justification for the implementation of an additional MI objective.

Weaknesses
- The related work section could be earlier in the paper, and possibly compared with LIPO in more detail. For example, what would the "domain knowledge" required by QD look like in the experimental settings described here?
- Figure 8 could include more context; is the goal to generate something that looks like the uniform distribution? Or are some choices clearly better than others in this environment? Essentially, interpreting this requires additional background knowledge about Overcooked and about what an ideal solution would look like.

**Summary Of The Paper:**

This paper introduces the LIPO (Learning Incompatible Policies) algorithm for generating diverse policies via a mutual information objective. This is then demonstrated empirically in two simple environments (a cooperative matrix game and a 2D navigation task), as well as in the Overcooked domain.

**Summary Of The Review:**

Overall, this is a strong paper with good results. It is likely to be useful to researchers seeking to design cooperative agents with diverse behaviors, as well as possibly for designing agents in non-cooperative and mixed cooperative-competitive settings.

---

> ### Author Response · Authors · 2022-11-13
> **Author response for reviewer PWGa**
>
> We thank the reviewer for commenting on the paper as being ___simple, powerful, reproducible, clearly written, and broadly applicable to multiple unknown settings where domain knowledge is difficult to obtain___. We are glad to hear that ___LIPO is likely to be useful to researchers seeking to design cooperative agents with diverse behaviors, as well as possibly for designing agents in non-cooperative and mixed cooperative-competitive settings___. We provide further clarification and address the existing concerns below.
>
> > The related work section could be earlier in the paper, and possibly compared with LIPO
> in more detail. For example, what would the "domain knowledge" required
> by QD look like in the experimental settings described here?
>
> We appreciate the suggestion of a deeper discussion on related methods. To give more context to the paper, as suggested by the reviewer, we added more details on the domain knowledge required by QD methods in the related work section. However, the added details are related to the environment used in the experiment; thus, we still put the related work section after the experiment section.
>
> > Figure 8 could include more context; is the goal to generate something that looks like the uniform distribution? Or are some choices clearly better than others in this environment? Essentially, interpreting this requires additional background knowledge about Overcooked and about what an ideal solution would look like.
>
> We thank the reviewer for the suggestion. The goal in this experiment is to learn a population that produces the most diverse recipe distribution. We added more information to clarify this part of the paper. We also added more details about the recipes in App. C.3.
>
> > It would be useful to compare LIPO further with MEP (Zhao et al. 2021) to discuss the qualitative/quantitative differences in policies that both obtain.
>
> We agree that an additional baseline would definitely provide additional insight. We will add more baselines in our future work. We hope that the reviewer is still satisfied with the provided set of baselines.
>
> > Line 1 of Appendix A: "proof" should be "prove”
>
> We thank the reviewer for noticing this mistake. We fixed the typo the in the revision.

---

### Official Review · Reviewer_CTS6 · 2022-10-27

**Confidence:** 4
**Correctness:** 4
**Technical Novelty And Significance:** 3
**Empirical Novelty And Significance:** 3
**Recommendation:** 8

**Clarity, Quality, Novelty And Reproducibility:**

The paper is well written, and has no obvious technical faults.  The approach to policy diversity is relatively novel, particularly the idea of explicitly training new policies to mis-coordinate with existing members of the population.  The authors provide their code, though this has note been checked.

**Strength And Weaknesses:**

The main strength of this paper is the intuitive nature of the LIPO approach, and the empirically demonstrated performance in both solution enumeration and as the basis for training generalist cooperative policies.

One potential weakness of LIPO, and related approaches, is the fact that they focus on enumerating the space of near-optimal solutions.  This means that generalist policies trained against these populations may perform poorly when paired with significantly sub-optimal partners.  In real-world settings, using LIPO-like methods could therefore lead to policies that are not robust to realistic partners who are unlikely to behave in a truly optimal way.

Another potential weakness is that there is no guarantee that the space of behaviors represented by a single policy parameterized by a latent state will represent a unique class of solutions.  Without the ability to recognize unique solutions, LIPO may not be able to recognize solutions that are unlikely to occur in a ad hoc setting (since other agents would not choose strategies that are likely to mis-coordinate).

**Summary Of The Paper:**

This paper presents the LIPO framework for learning diverse sets of policies for cooperative multi-agent reinforcement learning tasks.  Given an existing population consisting of tuples of joint policies, the LIPO objective maximizes the performance of a new joint policy, while penalizing it for performing well when paired with other policies within the population.  LIPO also, optionally, maximizes the diversity of the trajectories generated by an individual team of policies using policies parameterized by a randomly sampled latent variable, and an additional a mutual information objective.

Experiments demonstrate that LIPO is better than several recently developed alternatives at identifying the space of solutions to cooperative games, including simple matrix games, the cooperative particle environments, and the much more complex Overcooked! environment.  They demonstrate that LIPO is able to more reliably identify the full set of possible solutions, rather than just a subset of these solutions.  The also evaluate "generalist" policies trained against the population of policies generated with LIPO (or one of the baseline methods).  They demonstrate that policies trained against the LIPO population were significantly more robust (compared to the baseline methods) in terms of their performance when paired with members of a "test" populations.

**Summary Of The Review:**

The decision to accept is based on the empirical success of the proposed LIPO framework in training policies for ad hoc cooperative settings.  The method also represents something of a departure from existing approaches to learning diverse policies, which previously have focused on maximizing direct measures of the diversity of the policy space.

---

> ### Author Response · Authors · 2022-11-13
> **Author response for reviewer CTS6**
>
> We thank the reviewer for commenting on the paper as being ___intuitive, well written, and novel___. We are glad to hear that ___populations produced by LIPO is useful as a basis for training generalist agents___. We give our response to the reviewer below.
>
> > One potential weakness of LIPO, and related approaches, is the fact that they focus on enumerating the space of near-optimal solutions. This means that generalist policies trained against these populations may perform poorly when paired with significantly sub-optimal partners. In real-world settings, using LIPO-like methods could therefore lead to policies that are not robust to realistic partners who are unlikely to behave in a truly optimal way.
>
> We thank the reviewer for the comment. We agree with the reviewer that situations where test partners could behave sub-optimally are interesting settings. We added a discussion on this topic in the second paragraph of App. F. In short, Strouse et al. [1] show that augmenting the training population with past checkpoints (FCP) helps the trained generalist agent to effectively coordinate with sub-optimal, or even random, agents. Since LIPO and FCP are orthogonal, populations created by LIPO can be augmented in the same way as FCP.
>
> > Another potential weakness is that there is no guarantee that the space of behaviors represented by a single policy parameterized by a latent state will represent a unique class of solutions. Without the ability to recognize unique solutions, LIPO may not be able to recognize solutions that are unlikely to occur in a ad hoc setting (since other agents would not choose strategies that are likely to mis-coordinate).
>
> We thank the reviewer for pointing out potential weaknesses of LIPO. However, we don’t quite follow this specific comment. Could the reviewer elaborate on this? We are happy to discuss with the reviewer further.
>
> ### Reference
>
> [1] Strouse, D. J., et al. "Collaborating with humans without human data." *Advances in Neural Information Processing Systems*
>  34 (2021): 14502-14515.

---

> > ### Comment · Reviewer_CTS6 · 2022-11-15
> > **Reviewer response - solution classes**
> >
> > Apologies, I probably shouldn't have listed this as a weakness, as it is really a comment potential directions for future work.
> >
> > Previous work has examined the problem of identifying "symmetric" solutions to coordination problems (Hu 2020; Parker-Holder 2020).
> > If we imagine that a parametric family of policies (conditioned on the latent variable z) represent variations on a single solution, then we might exploit this approach to recognize such classes of solutions.  Classes which are incompatible with one another, and require prior coordination, may be less likely when paired with human partners.  It would be interesting to see whether this objective could be modified to identify families of *incompatible* policies that can be ignored by the generalist policy.
> >
> > Not a weakness of the current work though.
> >
> > **References:**
> >
> > Parker-Holder, Jack, et al. "Ridge rider: Finding diverse solutions by following eigenvectors of the hessian." Advances in Neural Information Processing Systems 33 (2020): 753-765.
> >
> > Hu, Hengyuan, et al. "“Other-Play” for Zero-Shot Coordination." International Conference on Machine Learning. PMLR, 2020.

---

> > > ### Author Response · Authors · 2022-11-16
> > > **Author response for the clarification by reviewer CTS6**
> > >
> > > We thank the reviewer for the clarification. We are glad that the reviewer envisions how LIPO can be useful for tackling another open question. The paper also includes a discussion related to this topic in App. F. In short, we agree that equivalent, but incompatible solutions might not be desirable in some environments. LIPO can be combined with other techniques, e.g., other-play (Hu et al., 2020) and equivariant coordinator (Muglich et al., 2022), to avoid learning arbitrary symmetry-breaking. We leave the study of the combination of LIPO and these techniques for future work.
> > >
> > >
> > > *We would highly appreciate if the reviewer could confirm that the concerns have been addressed. If so, we’d like to ask the reviewer to consider increasing the recommendation score. We are happy to discuss further if any additional questions arise.*
> > >
> > > ---
> > > **References:**
> > >
> > > [1] Hu, Hengyuan, et al. "“Other-Play” for Zero-Shot Coordination." International Conference on Machine Learning. PMLR, 2020.
> > >
> > > [2] Muglich, Darius, et al. "Equivariant Networks for Zero-Shot Coordination." arXiv preprint arXiv:2210.12124 (2022).

---

> > > > ### Comment · Reviewer_CTS6 · 2022-11-17
> > > > **Reviewer response**
> > > >
> > > > I believe you have addressed all of my concerns.  I still believe that my current score is the most appropriate one though taking into account both the technical correctness and potential impact of the work.

---

### Official Review · Reviewer_cwpF · 2022-10-28

**Confidence:** 4
**Correctness:** 4
**Technical Novelty And Significance:** 3
**Empirical Novelty And Significance:** 3
**Recommendation:** 8

**Clarity, Quality, Novelty And Reproducibility:**

What would be needed to extend these results to more than two players?

The results shown in Figure 9 are interesting, but is there a way to distill the key differences? I think the color scheme is hurting, as it isn't clear what cells are supposed to be compared.

**Strength And Weaknesses:**


The paper is very clearly written and well organized it is easy to read and follow the logic. The toy environments are highly diagnostic and give strong insights into why this algorithm outperformer baselines and other approaches. It was nice to see the algorithm tested both for its ability to generate diverse populations and show that those populations matter for behavior.

One key weakness is that the algorithms were all tested on new benchmarks created just for this manuscript. Since there are now many competing approaches to this kind of diversity generation, it would have been nice to see this algorithm outperform others on a task designed by others.

Further, I would like to see some ablation experiments to understand the role of the \lambda_MI in the Overcooked environment. It is further not clear in what contexts this term will be important.

**Summary Of The Paper:**

The authors develop a training procedure for two-player cooperative games that generate diverse policies. The idea is to generate policies that are incompatible when paired together and train agents with this loss function. This is tested in two toy environments and an Overcooked grid world. In all cases, LIPO finds diverse solutions (when possible) as well as diverse solutions.


**Summary Of The Review:**

The paper presents a new algorithm that is clearly motivated with theoretical and empirical support. I recommend the paper is accepted if my above comments can be addressed.

---

> ### Author Response · Authors · 2022-11-13
> **Author response for reviewer cwpF**
>
> We thank the reviewer for commenting on the paper as being ___clearly written, well organized, easy to understand, and clearly motivated with theoretical and empirical support___. We are glad to hear that ___the toy environments give strong insights into how LIPO works___. We provide further clarification and address the existing concerns of the reviewer below.
>
> > It would have been nice to see this algorithm outperform others on a task designed by others.
>
> We thank the reviewer for the suggestion. We also agree that applying LIPO to other environments would be a nice addition. The discussion regarding other well-known environments can be found at the third paragraph of App. G. We will explore the effectiveness of LIPO in more environments in future work.
>
> > Further, I would like to see some ablation experiments to understand the role of the $\lambda_{MI}$ in the Overcooked environment. It is further not clear in what contexts this term will be important.
>
> We thank the reviewer for the suggestion. We provided additional results without using the MI objective in multi-recipe Overcooked. Specifically, we included additional analysis in Sec. 4.5 and additional results recipe distribution in Fig. 8, recipe entropy in Table 1, and radar plot in Fig. 13. In summary, we find that LIPO populations with $\lambda_{MI}=0$ still, similar to $\lambda_{MI}=0.5$, consistently learn to use the hard-to-find Tomato & Carrot Salad recipe. However, the frequencies of Chopped Tomato and Chopped Carrot are lowered. This means that there are multiple joint policies that learn to complete the same recipe while being incompatible with each other. We suspect that using $\lambda_{MI}>0$ alleviates this problem by regularizing each joint policy to represent a policy with broader state-action coverage (e.g., learn multiple ways of completing a recipe), indirectly pushing other joint policies to use different recipes in order to be incompatible.
>
> > What would be needed to extend these results to more than two players?
>
> We have experimented with multi-player environments and achieved positive results with minimal algorithmic change. Specifically, $\pi_j$ is treated as all other agents, as stated in the footnote of page 2. We updated the pseudocode in App. D for applying LIPO to more-than-two-player environments.
>
> > The results shown in Figure 9 are interesting, but is there a way to distill the key differences? I think the color scheme is hurting, as it isn't clear what cells are supposed to be compared.
>
> We added more clarification explaining Fig. 9 in Sec. 4.6.

---

### Author Response · Authors · 2022-11-13
**Common response from the authors**

We thank all the reviewers for the overall positive sentiment towards the paper and their thoughtful feedbacks. We have made several updates to the paper as follows (the changes in the paper are highlighted in blue):

### List of changes

- Add a pointer to the extension to more than two player (Sec. 3.4)
- Add a pointer to the discussion on methods that utilize domain knowledge (Sec. 4)
- Clarification on the goal of the Overcooked experiment (Sec. 4.5)
- Clarification on the reference uniform distribution (Fig. 8)
- Additional discussion on using $\lambda_{MI}=0$ in multi-recipe Overcooked (Sec. 4.5)
- Additional figures with $\lambda_{MI}=0$ in multi-recipe Overcooked in Fig. 8, Tab. 1, App. J.2
- Clarification on the results of Fig.9 (Sec. 4.6)
- Additional discussion on the QD algorithm in the context of environments used in this paper (Sec. 5)
- Additional details on CMG (App. C.1)
- Additional details of the recipe reward in multi-recipe Overcooked (App. C.3)
- Update the pseudocode and discuss the training time when scaling to more than two players (App. D)

Some other minor changes, including figure resizing, had been made to obey the 9-page limit.

We hope that our responses and the changes made will resolve the questions and concerns raised by the reviewers. We would appreciate the reviewers checking the updates in the paper (highlighted in blue). We are happy to further discuss and clarify if any reviewers feel their comments are not addressed.

---

### Decision · Program_Chairs · 2023-01-20

**Decision:**

Accept: notable-top-25%

**Justification For Why Not Higher Score:**

Mostly empirical work, lack of complex environments for evaluation. Algorithmic innovation is limited (but effective!).

**Justification For Why Not Lower Score:**

Strong experimental results and analysis in the limited domains.

**Metareview: Summary, Strengths And Weaknesses:**

The authors propose and evaluate a simple yet effective method for generating diverse agents in Dec-POMDPs.
LIPO directly trains agents to maximise the reward when paired in self-play,  but minimise the reward when paired with other agents.
The authors also show that LIPO improves the cross-play performance when LIPO policies are matched with policies trained from unknown algorithms at test time.

While the paper does not offer a lot of theoretical or conceptual novelty the paper shines detailed empirical valuation. The only downside on the experimental side is the lack of complex environments -- prior work has shown that overcooked is a not a great testbed for studying coordination.

**Note From Pc:**

if the above contains the word "oral" or "spotlight" please see: "oral" presentation means -> notable-top-5% and "spotlight" means -> notable-top-25%. As stated in our emails, we are disassociating presentation type from AC recommendations